# Evaluating uncertainties in modelling the snow hydrology of the Fraser River Basin, British Columbia, Canada

Siraj Ul Islam and Stephen J. Déry

Environmental Science and Engineering Program,
University of Northern British Columbia,
3333 University Way, Prince George, BC, V2N 4Z9, Canada

*Correspondence to*: Stephen J. Déry (sdery@unbc.ca)

**Abstract.** This study evaluates predictive uncertainties in the snow hydrology of the Fraser River Basin (FRB) of British Columbia (BC), Canada, using the Variable Infiltration Capacity (VIC) model forced with several high-resolution gridded climate datasets. These datasets include the Canadian Precipitation Analysis and the thin-plate smoothing splines (ANUSPLIN), the North American Regional Reanalysis (NARR), University of Washington (UW) and Pacific Climate Impacts Consortium (PCIC) gridded products. Uncertainties are evaluated at different stages of the VIC implementation starting with the driving datasets, optimization of model parameters, and model calibration during cool and warm phases of the Pacific Decadal Oscillation (PDO).

The inter-comparison of the forcing datasets (precipitation and air temperature) and their VIC simulations (snow water equivalent (SWE) and runoff) reveal widespread differences over the FRB, especially in mountainous regions. The ANUSPLIN precipitation shows a considerable dry bias in the Rocky Mountains whereas the NARR winter air temperature is 2°C warmer than the other datasets over most of the FRB. In the VIC simulations, the elevation-dependent changes in the maximum SWE (maxSWE) are more prominent at higher elevations of the Rocky Mountains where the PCIC-VIC simulation accumulates too much SWE and ANUSPLIN-VIC yields an underestimation. Additionally, at each elevation range, the day of maxSWE varies 10 to 20 days between the VIC simulations. The

snow melting season begins early in the NARR-VIC simulation whereas the PCIC-VIC simulation delays the melting indicating seasonal uncertainty in SWE simulations. When compared with the observed runoff for the Fraser River main stem at Hope, BC, the ANUSPLIN-VIC simulation shows considerable underestimation of runoff throughout the water year owing to reduced precipitation in the

5 ANUSPLIN forcing dataset. The NARR-VIC simulation yields more winter and spring runoff and earlier decline of flows in summer due to a nearly 15-day earlier onset of the FRB springtime snowmelt.

Analysis of the parametric uncertainty in the VIC calibration process shows that the choice of the initial parameter range plays a crucial role in defining the model hydrological response for the FRB. Furthermore, the VIC calibration process is biased toward cool and warm phases of the PDO and the

10 choice of proper calibration and validation time periods is important for the experimental setup. Overall the VIC hydrological response is prominently influenced by the uncertainties involved in the forcing datasets rather than those in its parameter optimization and experimental setups.

**Keywords:** Uncertainties, hydrological modelling, climate datasets, runoff, calibration, Fraser River

# 1 Introduction

While advances in computational power and ongoing developments in hydrological modelling have increased the reliability of hydrologic simulations, the issue of adequately addressing the associated uncertainty remains challenging (Liu and Gupta, 2007). There is a growing need for proper estimation of uncertainties associated with hydrological models and the observations required to drive and evaluate their outputs. Hydrological simulations of snow processes and related hydrology depend critically on the input climate forcing datasets, particularly the precipitation and air temperature (Reed et al., 2004; Mote et al., 2005; Tobin et al., 2011). Both of these input forcings regulate the quantity and phase of modelled precipitation and affect the response of simulated snow accumulation and runoff. The model results therefore rely heavily on the quality of these forcings as the uncertainty (measurement errors, etc.) in such data will propagate through all hydrological processes during simulations (Wagener and Gupta, 2005; Anderson et al., 2007; Tapiador et al., 2012). Studies such as Essou et al. (2016a) compared hydrological simulations of different observed datasets over the continental United States (US). They reported that there are significant differences between the datasets, although all the datasets were essentially interpolated from almost the same climate databases. Furthermore, Essou et al. (2016b) compared the hydrological response of three reanalysis datasets over the US and found precipitation biases in all reanalyses, especially in summer and winter in the southeastern US. The uncertainties in hydrological simulations also arise from the model parameters, its structure and in the objective function and the calibration variable that is used for model calibration. Hence the reliability of input forcings along with the capability of the hydrological model and the experimental setup ultimately determine the fate of hydrological variables essential for water resource management.

Several observed gridded climate datasets of precipitation and air temperature (Mesinger et al., 2006; Hopkinson et al., 2011), based on available observational data, post-processing techniques and, in some cases, climate modelling, are currently available over the Canadian landmass to facilitate climate and hydrological simulations. These datasets provide long term gridded precipitation and air temperature records on hourly and daily bases making them especially useful for hydrological simulations, particularly over areas where in situ station densities are low. However, these datasets, being spatially interpolated or assimilated to gridcells, rely mainly on the spatial density of the observational network, which is often quite low in mountainous regions (Rinke et al., 2004). Observational data incorporated into gridded datasets may also contain measurement errors and missing records that translate into the data interpolation and contribute to the overall uncertainty in gridded data products. Such uncertainties are assessed in many studies focusing on the forcing data (Horton et al., 2006; Graham et al., 2007; Kay et al., 2009; Eum et al., 2014).

The quality of hydrological modelling depends on how well a model simulates the regional detail and topographic characteristics of the region, especially in mountainous regions. However, most mountainous regions exhibit higher errors in gridded datasets because they are usually based on an uneven number of stations that are mostly located at lower elevations (Eum et al., 2012). This is true for most large basins in western Canada that exhibit highly variable elevation ranges and strong climatological heterogeneity. One such large basin is British Columbia's (BC's) Fraser River Basin (FRB), which is vital for Canada's environment, economy and cultural identity. Its mountainous snowpack serves as a natural reservoir for cold-season precipitation, providing snowmelt driven flows in summer. Evaluating uncertainties in modelling the FRB's hydrology is crucial for informed decision-

making and water resources management. This includes the communication of the uncertainties, propagated into the model predictions, in an appropriate manner to decision makers or stakeholders, thereby allowing confidence in the model results.

Although the currently available gridded datasets (reanalysis and interpolated) over the FRB are derived from observational stations using various interpolation and assimilation techniques, they may still have systematic biases because of their grid resolution, the density of the surface station network used for data assimilation, and the topographic characteristics of the FRB. In the FRB, 23% of the basin exceeds 1500 m in elevation whereas roughly 5% of the in situ meteorological stations surpasses this elevation (Shrestha et al., 2012). Such mismatch between station densities at different elevations makes the precipitation interpolation at higher elevations excessively influenced from the lower elevation stations (Stahl et al., 2006; Rodenhuis et al., 2009; Neilsen et al., 2010). Therefore, despite extensive implementation of hydrologic modelling with single observed forcings (e.g. Shrestha et al., 2012; Kang et al., 2014, 2016), evaluation of the uncertainties in forcing datasets remains a critical and challenging issue for the FRB. As such, the first step is to evaluate available observation-based forcing datasets for their suitability to be used in hydrological modelling over the FRB.

In Canada, numerous studies have assessed the performance of hydrologic simulations driven by only one particular driving dataset (Pietroniro et al., 2006; Choi et al., 2009; Bennett et al., 2012; Kang et al., 2014). Sabarly et al. (2016) used four reanalysis datasets to assess the terrestrial branch of the water cycle over Quebec with satisfactory results over 1979-2008. Eum et al. (2014) recently compared hydrological simulations driven by several high-resolution gridded climate datasets over western Canada's Athabasca watershed and found significant differences across the simulations. While BC's

snowpacks and hydrology are well studied in the literature (Danard and Murty, 1994; Choi et al., 2010; Thorne and Woo, 2011; Déry et al., 2012; Shrestha et al., 2012; Kang et al., 2014, 2016; Islam et al., 2017; Trubilowicz et al., 2016), detailed inter-comparisons of available observational forcing in terms of their hydrological response is not thoroughly analysed, particularly over the FRB's complex topography. In this study, we therefore investigate the simulated hydrological response of uncertainties associated with air temperature and precipitation forcing on the FRB's mountainous snowpack and runoff. To achieve this, four forcing datasets, namely the Canadian Precipitation Analysis and the thin-plate smoothing splines (ANUSPLIN hereafter; Hopkinson et al., 2011), the North American Regional Reanalysis (NARR hereafter; Mesinger et al., 2006), University of Washington (UW hereafter; Shi et al., 2013) and Pacific Climate Impacts Consortium (PCIC hereafter; Shrestha et al., 2012) gridded products are applied to the FRB. These datasets are explored across three different regions and multiple elevation ranges. The PCIC and UW datasets are used by Shrestha et al. (2012) and Kang et al. (2014, 2016), respectively to drive the VIC hydrological model over the FRB whereas the NARR and ANUSPLIN datasets are not yet evaluated over this region. However, the NARR dataset is used in studies focusing on other regions of Canada (Woo and Thorne, 2006; Choi et al, 2009; Ainslie and Jackson, 2010; Eum et al., 2014; Trubilowicz et al., 2016). To our knowledge, this is the first comprehensive study that collectively examines the spatial and elevation dependent hydrological response of these datasets for the FRB.

Along with forcing datasets, many studies have focused their attention either on model structure (Wilby and Harris, 2006; Jiang et al., 2007; Poulin et al., 2011; Velazquez et al., 2013) or on calibration parameters (Teutschbein et al., 2011; Bennett et al., 2012). Arsenault et al. (2014) estimated the

uncertainty due to parameter set selection using the hydrological model over several basins in Quebec. They showed that parameter set selection can play an important role in model implementation and predicted flows. For parameter uncertainty, a hydrological model can have many equivalent local optima within a realistic parameter space (Poulin et al., 2011). Therefore, several different parameter sets may be available for the same "optimal" measure of efficiency during the optimization process (i.e. parameter non-uniqueness; Beven, 2006). Here we evaluate the parameter uncertainties involved in the model calibration process, i.e. calibration optimizer sensitivity to parameter initial limits. Moreover we focus on another unique aspect of modelling uncertainty related to the selection of time periods for model calibration and validation under changing climatic conditions on decadal time scales. Studies such Klemeš (1986) and Seiller et al. (2012) highlighted the issue of calibration and validation of hydrological modelling under different climatological conditions. In this study, we estimate the hydrological model sensitivity to different climatological conditions by focusing on the FRB's air temperature and precipitation teleconnections with cool and warm phases of the Pacific Decadal Oscillation (PDO).

Overall, the main goals of this study are: (i) to compare and identify the most reliable available gridded forcing datasets for hydrological simulations over the FRB; (ii) to evaluate hydrological modelling responses of different driving datasets over a range of FRB elevations; (iii) to assess the uncertainty involved in the model calibration process by focusing on the optimizer used for parameter optimization; and (iv) to evaluate the calibration process under changing climatic conditions. To achieve these four objectives, the macroscale Variable Infiltration Capacity (VIC) hydrological model (Liang et al., 1994, 1996) is used as the simulation tool. The VIC model conserves surface water and energy balances for

large-scale watersheds such as the FRB (Cherkauer et al., 2003). It has been successfully implemented, calibrated and evaluated over the FRB (Shrestha et al., 2012; Kang et al., 2014; Islam et al., 2017).

The remainder of this paper is structured as follows. Section 2 discusses the FRB, the driving datasets, the VIC model and experimental setup. Section 3 describes the forcings inter-comparison, hydrological simulations, parameter sensitivity and uncertainly related to the PDO. Section 4 summarizes and concludes this study.

## 2. Study Area, Model and Methodology

### 2.1 Fraser River Basin (FRB)

The FRB is one of the largest basins of western North America spanning 240,000 km$^2$ of diverse landscapes with elevations varying from sea level to 3954 m above sea level at Mt. Robson, its tallest peak (Benke and Cushing, 2005). It covers the mountainous terrain of the Coast and Rocky Mountains along with dry central plateaus (Fig. 1). The FRB's headwaters are in the Rocky Mountains with its major tributaries being the Stuart, Nechako, Quesnel, Chilcotin, Thompson, and Harrison Rivers. The Fraser River runs 1400 km through the whole basin before reaching Hope, BC, where it veers westward to drain into the Salish Sea and the Strait of Georgia at Vancouver, BC (Benke and Cushing, 2005; Schnorbus et al., 2010).

In winter, considerable amounts of snow usually accumulate at higher elevations, except in coastal areas. In late spring and early summer, snowmelt from higher elevations induce peak flows in the main stem of the Fraser River and its many tributaries (Moore and Wondzell, 2005), which rapidly decline in

late summer following the depletion of snowmelt. Owing to its complex mountainous ranges, the FRB's hydrologic response varies considerably across the basin, differentiating it into snow-dominant, hybrid (rain and snow), or rain-dominant regimes (Wade et al., 2001). Glaciers cover only 1.5% of the FRB (Shrestha et al., 2012) and provide only a modest contribution to streamflow, primarily in late summer (August/early September).

## 2.2 Datasets

Along with recent developments in hydrological models, several observation-based gridded datasets are now available to drive the models such as ANUSPLIN, NARR, UW and PCIC. These meteorological forcing datasets are developed using high-resolution, state-of-the-art data interpolation and (for NARR only) assimilation techniques. This is to improve the quality of forcing data to analyse a model's hydrological response over any particular basin.

The ANUSPLIN dataset, developed by Natural Resources Canada (NRCan), contains gridded data of daily maximum and minimum air temperature (°C), and total daily precipitation (mm) for the Canadian landmass south of 60°N at ~10 km resolution (NRCan, 2014). This Canadian dataset uses a trivariate thin-plate smoothing spline technique referred to as ANUSPLIN (Hutchinson et al., 2009) with recent modifications (Hopkinson et al., 2011). Eum et al. (2014) used the ANUSPLIN dataset for hydrological modelling over Alberta's Athabasca watershed and reported underestimations in simulated runoff, owing to a dry bias in ANUSPLIN precipitation.

NARR was developed at 32 km spatial and 3-hourly temporal resolution to improve the National Centers for Environmental Prediction (NCEP)/National Center for Atmospheric Research (NCAR)

global reanalysis data by employing the Eta Data Assimilation system for the North American domain for the period from 1979 to the current year. The interannual variability of the NARR seasonal precipitation and accuracy of its temperature and winds are found superior to earlier versions of the NCEP/NCAR reanalysis datasets (Mesinger et al., 2006; Nigam and Ruiz-Barradas, 2006). Choi et al.

(2009) investigated the applicability of air temperature and precipitation data from NARR for hydrological modelling of selected watersheds in northern Manitoba. They found that NARR air temperature and precipitation data are in much better agreement with observations than the NCEP–NCAR Global Reanalysis-1 dataset (Kalnay et al., 1996; Kistler et al., 2001). Woo and Thorne (2006) used air temperature and precipitation data from two global reanalysis datasets and from NARR as input

to a hydrological model for the Liard River Basin in western subarctic Canada and reported significant improvement in its hydrological simulations. NARR output has also been used in regional water budget calculations (Luo et al., 2007; Ruane, 2010; Sheffield et al., 2012). Choi et al. (2009) and Keshta and Elshorbagy (2011) reported that NARR output is suitable for hydrologic modelling, especially when other observations are unavailable. However, they focused on the Canadian Prairies, where the

topography is not complex.

The UW dataset of daily precipitation, maximum and minimum air temperature, and average wind speed are based on the extended gridded UW dataset (Shi et al., 2013; Adam et al., 2006; Adam and Lettenmaier, 2008). Monthly precipitation originates from the University of Delaware observed land surface precipitation product (Matsuura and Willmott, 2009), which was converted to daily data using

the high temporal precipitation dataset from Sheffield et al. (2006). To improve the precipitation estimates, the monthly data were adjusted to account for gauge undercatch by using the methods

outlined by Adam and Lettenmaier (2008). Such adjustment is important since gauge-based precipitation measurements may underestimate solid precipitation in winter by 10%−50% (Adam and Lettenmaier 2003). Daily wind speeds are extracted from the NCEP/NCAR reanalysis datasets (Kalnay et al., 1996).

The PCIC dataset of precipitation, maximum and minimum temperature, and wind speed was derived primarily from Environment and Climate Change Canada (ECCC) climate station observations, with additional inputs from the United States Co-operative Station Network, the BC Ministry of Forests, Lands and Natural Resource Operations, the BC Ministry of Environment's Automated Snow Pillow network, and BC Hydro's climate network (Schnorbus et al., 2011; Shrestha et al., 2012). These data
are available at ~6 km resolution and were corrected for point precipitation biases and elevation effects (Schnorbus et al., 2011).

  The ANUSPLIN, NARR, UW and PCIC datasets are available at 10 km, 32 km, 25 km and 6 km spatial resolution, respectively, and at a daily time scale. To facilitate comparison, the ANUSPLIN, NARR and PCIC datasets were regridded to 25 km resolution using bilinear interpolation to match the
scale of the current VIC implementation. The NARR (32 km) dataset was interpolated from coarse resolution curvilinear grids to slightly higher (25 km) resolution rectilinear grids. On the other hand, both the PCIC (6 km) and ANSUPLIN (10 km) datasets were interpolated to a coarser resolution (25 km). The elevation correction, which is important when interpolating from coarser to higher spatial resolutions (Dodson and Marks, 1997), was not used to correct the orographic effects for the NARR
dataset. Interpolating the NARR dataset from a 32 km to a 25 km spatial resolution induces negligible elevation dependent uncertainties as elevation changes remain below ±20% in the FRB, with most of

the grid cells having nearly no difference in orography. Thus the relationship of atmospheric variables such as air temperature with elevation remains nearly identical at both resolutions.

Daily wind speeds, a required VIC input variable, are not available for the ANUSPLIN dataset. We therefore used the PCIC based wind speeds in the ANUSPLIN driven VIC simulations. The PCIC wind speeds are sourced from the Environment and Climate Change Canada station product (Schnorbus et al., 2011).

To calibrate and validate the VIC model simulated flows, we used daily streamflow data from ECCC's Hydrometric Dataset (HYDAT; Water Survey of Canada, 2014). These data were extracted and compiled into a comprehensive streamflow dataset for the FRB spanning 1911–2010 (Déry et al., 2012). In addition, we compared the simulated SWE with observations from the BC River Forecast Centre's network of snow pillow sites (BC Ministry of Forests, Lands and Natural Resource Operations, 2014). The snow pillow stations record the mass of the accumulated snowpack (SWE) on a daily basis. Based on the availability of data, we used SWE observations from four sites located at Yellowhead (ID: 1A01P) and McBride (ID: 1A02P) in the upper Fraser and at Mission Ridge (ID: 1C18P) and Boss Mountain Mine (ID: 1C20P) in the middle Fraser. Due to data availability, we used the 1996-2006 time period for the Yellowhead, Mission Ridge and Boss Mountain Mine snow pillows and 1980-1986 for the McBride location. Detailed information about these sites is available in Kang et al. (2014) and Déry et al. (2014).

## 2.3 Variable Infiltration Capacity (VIC) Model

The VIC model resolves energy and water balances and therefore requires a large number of parameters, including soil, vegetation, elevation, and daily meteorological forcings, at each gridcell. To evaluate hydrological responses over complex terrain, the model simulates the subgrid variability in topography and precipitation by dividing each gridcell into a number of snow elevation bands (Nijssen et al., 2001a). The model utilizes a mosaic-type representation by partitioning elevation bands into a number of topographic tiles that are based on high-resolution spatial elevations and fractional area. The snow model embedded in the VIC model is then applied to each elevation tile separately (Gao et al., 2009).

The VIC model is widely used in many hydrological applications including water availability estimation and climate change impacts assessment in North America (Maurer et al., 2002; Christensen and Lettenmaier, 2007; Adam et al., 2009; Cuo et al., 2009; Elsner et al., 2010; Gao et al., 2010; Wen et al., 2011; Oubeidillah et al., 2014) and around the world (Nijssen et al., 2001a,b; Haddeland et al., 2007; Zhou et al., 2016). It is also commonly used to simulate hydrologic responses in snowmelt-dominated basins (Christensen and Lettenmaier, 2007; Hidalgo et al., 2009; Cherkauer and Sinha, 2010; Schnorbus et al., 2011).

### 2.3.1 The VIC Implementation

The VIC model, as set up by Kang et al. (2014) and Islam et al. (2017) for the FRB, is employed for evaluating the model's ability to simulate the FRB's hydrological response when driven by different observational forcings. The model was previously applied to the FRB to investigate its observed and

projected changes in snowpacks and runoff. In this study, we performed model integrations over the entire FRB using gridcells spanning 48°– 55°N and 119°–131°W. The model is configured at 0.25° spatial resolution using a daily time step, three soil layer depths and ten vertical snow elevation bands. Once an individual VIC simulation is completed, the runoff for the basin is extracted at an outlet point

of the given sub-basin, using an external routing model that simulated a channel network (adapted from Wu et al., 2011) with several nodes (Lohmann et al., 1996; 1998a, b). Streamflow is converted to areal runoff by dividing it by the corresponding sub-basin area. Daily runoff at the outlet cell is integrated over time to obtain total water year runoff for a selected basin. Other than the calibration parameters, the soil and vegetation parameters, leaf area index (LAI) and albedo data are kept identical as per the

Kang et al. (2014) VIC model implementation to the FRB .

**2.3.2 Calibration**

To explore the feasible parameter space, we used the University of Arizona multi-objective complex evolution (MOCOM-UA) optimizer for the VIC calibration process (Yapo et al., 1998; Shi et al., 2008). MOCOM-UA searches a set of VIC input parameters using the population method to maximize the

15 Nash–Sutcliffe efficiency (NSE) coefficient (Nash and Sutcliffe, 1970) between observed and simulated runoff. Six soil parameters are used in the optimization process, i.e. b_infilt (a parameter of the variable infiltration curve), Dsmax (the maximum velocity of base flow for each gridcell), Ws (the fraction of maximum soil moisture where nonlinear base flow occurs), D2 and D3 (the depths of the second and third soil layers), and Ds (the fraction of the Dsmax parameter at which nonlinear baseflow occurs).

These calibration parameters were selected based on the manual calibration experience from previous

studies by Nijssen et al. (1997), Su et al. (2005), Shi et al. (2008), Kang et al. (2014, 2016) and Islam et al. (2017). VIC is a physically based hydrologic model that has many (about 20, depending on how the term "parameter" is defined) parameters that must be specified. However, the usual implementation approach involves the calibration of only these six soil parameters. Such parameters have the largest effects on the hydrograph shape and are the most sensitive parameters in the water balance components (Nijssen et al., 1997; Su et al., 2005). These parameters must be estimated from observations, via a trial and error procedure that leads to an acceptable match of simulated discharge with observations.

For the snow calibration, the value of thresholds for maximum (at which snow can fall) and minimum (at which rain can fall) air temperature were fixed as 0.5°C and -0.5°C, respectively. These values were adjusted based on the region's climatology and were kept constant for all simulations in the global control file. Parameters related to the snow albedo were adjusted using the traditional VIC algorithm based on the US Army Corps of Engineers empirical snow albedo decay curves for transitions from snow accumulation to ablation.

Final values of these six calibrated parameters were estimated for each forcing dataset by a number of simulation iterations minimizing the difference between the simulated and observed monthly flow.

While the MOCOM-UA automated optimization process utilizes monthly streamflow during calibration, we evaluated the overall model performance on daily time scales using NSE and correlation performance metrics.

The VIC model calibration is applied to the Fraser River's main stem at Hope, BC and the FRB's major sub-basins, namely the Upper Fraser at Shelley (UF), Stuart (SU), Nautley (NA), Quesnel (QU), Chilko

(CH) and Thompson-Nicola (TN) basins (Fig. 1a and Supplementary Table 1). These sub-basins contribute 75% of the annual observed Fraser River discharge at Hope, BC with the largest contributions from the TN, UF and QU sub-basins (Déry et al., 2012).

### 2.3.3 Experiments

A series of different VIC experiments was performed to (i) compare the VIC model's response when driven by different forcings, (ii) evaluate the uncertainties related to the VIC optimizer, and (iii) investigate the effect of PDO teleconnections on the VIC calibration and validation time periods. For objective (i), we used all the four datasets to run VIC simulations to facilitate detailed comparison of different datasets and their hydrological response. In objectives (ii) and (iii), rather than the inter-

comparison of datasets, our goal is to evaluate the uncertainties in the model implementation particularly in its calibration process. We therefore only used the UW dataset to force the VIC model as this dataset along with our VIC model implementation is examined extensively over the FRB in Kang et al. (2014) and (2016). The experiments are categorized as follows:

1) **Inter-comparison runs:** The VIC model was driven by each forcing dataset for 28 years (1979 to

15 2006) with 1979-1990 as the calibration period and 1991-2006 as the validation period using the MOCOM-UA optimizer (Table 1). The VIC simulations driven by ANUSPLIN, UW and PCIC forcings are initiated five years prior to the year 1979 to allow model spin-up time. Since NARR is not available until 1979, its VIC simulations were recursively looped for five years using the year 1979 as the forcing data. After calibration, the model validation runs were initialized with five

different state files to produce five ensemble members. The ANUSPLIN, NARR, UW and PCIC

driven ensemble mean VIC simulations are referred to as ANUSPLIN-VIC, NARR-VIC, UW-VIC and PCIC-VIC, respectively. These ensemble simulations were run for the whole FRB and its UF, SU, NA, QU, CH and TN sub-basins.

2) **Optimizer uncertainty runs:** Here we only used the UW forcing data for VIC model simulations to investigate the uncertainties in the model calibration process for the 1979-1990 time period. Our primary goal is to evaluate optimizer sensitivity to a unique set of parameter limits. We want to see how the MOCOM optimizer results in different optimized parameters and change the overall simulated hydrograph in the calibration process. We performed the optimization of six soil parameters, i.e. b_infilt, Dsmax, Ws, D2, D3 and Ds in five experimental setups using different initial ranges of parameter limits. The VIC calibration experiments (OPT1, OPT3, OPT4 and OPT5) were run using four narrow ranges selected from the maximum limits of calibration parameters. The same experiment is then run with maximum limits of the calibration parameters (OPT2). Calibration parameters, their initial ranges and final optimized values for all the experiments are given in Table 3. The OPT1, OPT2, OPT3, OPT4 and OPT5 simulations were run over the whole FRB only.

3) **PDO uncertainty runs:** We used the UW dataset to drive long term (1950-2006) VIC simulations. This is to capture the decadal variability of cool and warm phases of the PDO. Five different experiments, namely PDO1, PDO2, PDO3, PDO4 and PDO5 were performed with calibration periods of 1981-1990, 1956-1965, 1967-1976, 1977-1987 and 1991-2001 and with corresponding validation periods of 1991-2001, 1966-1976, 1977-1987, 1967-1976 and 1981-1990, respectively (Table 4). Each time period was selected to capture cool or warm PDO phases, i.e. its cool (1956-1965 and 1967-1976) and warm (1981-1990, 1991-2001 and 1977-1987) phases. For each

calibration experiment in one particular phase of the PDO, the MOCOM-UA was used to optimize calibration parameters. The NSE was calculated for the calibration and validation periods using the daily observed streamflow data for the Fraser River at Hope. All PDO simulations were run over the whole FRB only.

## 2.4 Analysis Strategy

The analyses were performed for three FRB hydro-climatic regimes: the Interior Plateau, the Rocky Mountains and the Coast Mountains (Moore, 1991). These three regions were chosen given their distinct physiography and hydro-climatic conditions. The gridcell partitioning of these three regions and their elevations are shown in Fig. 1b. Results in this study mainly focused on the Fraser River main stem at Hope, BC since it covers 94% of the basin's drainage area and has a continuous streamflow record over the study periods. However, the inter-comparison runs were also compared over the FRB's major sub-basins. The total runoff was calculated using the sum of baseflow and runoff. Seasonal variations were assessed by averaging Dec-Jan-Feb (DJF), Mar-Apr-May (MAM), Jun-Jul-Aug (JJA) and Sep-Oct-Nov (SON) months for winter, spring, summer and autumn, respectively.

In the SWE analysis, the snowmelt was calculated by taking the difference between maximum and minimum SWE over the water year (1 October to 30 September of the following calendar year). The corresponding day of the water year having maximum SWE (maxSWE) is referred to as maxSWE-day.

Although glacier dynamics are not included in the VIC model physics, the model produces a perennial snowpack in several gridcells in its output. We compared those cells to Baseline Thematic Mapping (BTM) and found that the glaciating cells match the location of observed glaciers. We therefore masked

those gridcell in the SWE analysis considering that the effects of glaciers may not change our results significantly due to the ~25 km model grid cell resolution (625 km$^2$ area per grid cell) used in this study.

The Mann–Kendall test (Mann, 1945; Kendall, 1970) was used to estimate monotonic trends in the input forcing data and the simulated hydrological variables. This non-parametric trend test has been used in several other studies to detect changing hydrological regimes (Lettenmaier et al., 1994; Ziegler et al., 2003; Déry et al., 2005, 2016; Kang et al., 2014). Trends were considered to be statistically-significant when $p < 0.05$ with a two-tailed test.

## 3 Results and Discussion

We first examine the ANUSPLIN, NARR, UW and PCIC datasets to investigate how substantial are the differences in precipitation and air temperature at several temporal and spatial scales across the FRB and its sub-regions. The VIC simulations, driven by these forcing datasets, are then discussed to evaluate uncertainties in simulated SWE and runoff. This is followed by the discussion of uncertainty in the VIC calibration process.

### 3.1 Forcings Datasets Inter-comparison

The daily mean air temperature of ANUSPLIN, NARR, UW and PCIC datasets remains below 0°C from November to March and rises above 0°C in early spring over all three FRB sub-regions (Fig. 2). While the inter-datasets seasonal variability of air temperature is quite similar, the winter in NARR is ~2°C warmer compared to the remaining datasets. The grid scale seasonal differences (PCIC minus ANUSPLIN, NARR and UW) of mean air temperature spatially quantify the inter-datasets

disagreements (Supplementary Fig. 1). While the PCIC-ANUSPLIN and PCIC-UW differences are within ±1°C, the PCIC-NARR difference exceeds 2°C over most of the FRB in DJF and SON, revealing NARR air temperatures to be quite warmer than in the PCIC dataset.

The magnitudes of daily mean precipitation vary markedly amongst datasets. Winter precipitation, which begins in November and persists until April, shows greater inter-datasets differences, particularly over the Rocky and Coast Mountains. Compared to the PCIC and UW datasets, the ANUSPLIN precipitation is underestimated in all three regions with nearly 2.0 mm day$^{-1}$ to 5.0 mm day$^{-1}$ differences in the Rocky and Coast Mountains, respectively. This underestimation is more evident in the PCIC-ANUSPLIN spatial difference revealing up to 5 mm day$^{-1}$ difference over the mountainous regions (Supplementary Fig. 2). The precipitation differences in the Interior Plateau approach zero for all datasets. The maximum intraseasonal variability arises in the Coast Mountains ranging from 10.0 mm day$^{-1}$ of precipitation in winter and nearly zero in summer. The range of inter-datasets spread for peak precipitation varies from 5.0 mm day$^{-1}$ to 10.0 mm day$^{-1}$ during winter for the Coast Mountains. Precipitation in the Coast Mountains is more variable due to its proximity to the Pacific Ocean where the interaction between steep elevations and storm track positions is quite complex. In the Coast Mountains, the NARR precipitation is underestimated and is comparable to ANUSPLIN.

The underestimation of the ANUSPLIN mountainous precipitation is probably due the thin plate smoothing spline surface fitting method used in its preparation. For NARR, air temperature and precipitation uncertainties may have been induced by the climate model used to assimilate and produce the reanalysis product.

## 3.2 Hydrological Simulations

The ANUSPLIN-VIC, NARR-VIC, UW-VIC and PCIC-VIC simulation performance was evaluated using the NSE and correlation coefficients by calibrating and validating against observed daily streamflow for the Fraser River at Hope (Table 2). The NSE scores are much higher for the PCIC-VIC and UW-VIC simulations compared to the ANUSPLIN-VIC and NARR-VIC. The lower NSE score in the ANUSPLIN-VIC simulation reflects a dry precipitation bias in the ANUSPLIN dataset. As the model configuration, resolution, and soil data were identical for all VIC simulations, different NSE values reveal uncertainty associated only with each observational forcing dataset. Despite the low NSE score of the ANUSPLIN-VIC simulation, the correlation coefficient is significantly high. The bias in the simulated streamflow is contributing to the lower NSE coefficient whereas the phase of seasonal flow is quite similar to the observed flow in the ANUSPLIN-VIC simulation. There may be additional sources of uncertainty due to the method used to assess simulation accuracy. For example, instead of using NSE, other model evaluation metrics such as the Kling-Gupta Efficiency (KGE) coefficient (Gupta et al., 2009) may produce different levels of model accuracy.

The ANUSPLIN-VIC, NARR-VIC, UW-VIC and PCIC-VIC simulated SWE and snowmelt, areally-averaged over the FRB's three sub-regions, show similar seasonal variability but considerably different magnitudes, especially over mountainous regions. Figure 3a shows these differences for the Rocky Mountains revealing the range of peak SWE from 400 mm for ANUSPLIN to >600 mm for PCIC. The dry bias in ANUSPLIN precipitation forcing induces lower SWE magnitudes in the ANUSPLIN-VIC simulation. The lower SWE in the NARR-VIC simulation is probably due to the warmer air temperature during winter and spring (Fig. 2b). Winter temperatures being warmer in the NARR dataset may alter

the phase of precipitation partitioning with more rainfall than snowfall, and hence less SWE in the NARR-VIC simulation. Such differences in SWE are reflected in the associated snowmelt (Fig. 3b) where the NARR-VIC simulation shows earlier snowmelt. This is further investigated by VIC sensitivity experiments and is discussed later in the text. Grid-scale differences in simulated SWE (Fig. 4) and runoff (Supplementary Fig. 3) arise most notably over the mountainous regions. In the interior FRB, the simulation differences between PCIC-VIC and ANUSPLIN-VIC mean SWE are within a 10 mm range whereas such differences exceed 50 mm to 100 mm for the NARR-VIC and UW-VIC simulations.

In the FRB's mountainous regions, the VIC model can lead to inaccurate snowpack estimates if the elevation dependence on snow accumulation and ablation is not modelled properly. As mentioned in section 2.3, we used ten elevation bands in our VIC implementation so that each band's mean elevation was used to lapse the gridcell average air temperature and precipitation to produce more reliable estimates. We clustered the elevation distribution within 10 bands into different elevation ranges. This allowed in-depth analysis of the elevation dependent variation of mean SWE that is of particular importance for the Rocky and Coast Mountains regions of the FRB. We examined the magnitude of maxSWE and corresponding maxSWE-day of the water year between all simulations and elevation ranges (Fig. 5). The difference in maxSWE between all VIC simulations increases with elevation, particularly the Rocky Mountains where higher elevations (>1400 m) show large disagreement between simulated maxSWE (Fig. 5a). In the Interior Plateau, the NARR-VIC simulated maxSWE exceeds 300 mm whereas all other simulations are within 200 mm. The maxSWE elevation dependent variation is quite complex in the Coast Mountains. However the simulation differences at elevations >1400 m are

smaller compared to the lower elevations below 1000 m. Apart from maxSWE magnitude, the maxSWE-day variation differs considerably across the VIC simulations. Generally, the maxSWE-day varies by nearly two months between lower and higher elevations as snow onset occurs later in autumn. While the maxSWE-day variation is quite complex within each elevation range, the NARR-VIC maxSWE-day is earliest whereas PCIC-VIC delays the snow accumulation over the 600-2000 m elevation range in the Rocky Mountains. There is nearly 20 days of simulated variation in maxSWE-day at the Rocky Mountains highest elevation range. Such variation highlights the uncertainties in seasonality of the VIC simulated snowpacks. For the Interior Plateau and the Coast Mountains, no consistent pattern of maxSWE-day variation exists for any particular simulation.

**3.2.1 Comparison of observed versus simulated SWE**

As mentioned earlier, all gridded climate forcing datasets are based on station observations. The density of stations in the FRB's mountainous regions remains quite low and therefore induces higher uncertainties in the observational gridded products. It is important to quantify the spatial discrepancy between the simulated (0.25° gridcell) and observed (snow pillow station dataset) SWE that may lead to an uncertainty in snow estimations by models (Elder et al., 1991; Tong et al., 2010). We used observed SWE from BC snow pillow sites and the VIC simulated SWE data over the same elevation and overlapping continuous time periods at four different locations in the upper and middle Fraser where a high volume of SWE accumulates seasonally.

The daily time series of VIC simulated SWE (Supplementary Fig. 4) follows the observed interannual variability in snow accumulation but with considerable differences across simulations. The PCIC-VIC

simulation accumulates too much SWE compared to observations in the gridcell corresponding to the Yellowhead location. This overestimation is further explored for this site by expanding the time series back to 1979 (not shown), which reveals issues with PCIC precipitation data only during 1996-2004 with considerable above normal anomalies at Yellowhead. While ANUSPLIN-VIC shows lower SWE amounts, the NARR-VIC and UW-VIC simulations reproduce the observed variation quite reasonably for Yellowhead. For McBride, all simulations are more or less comparable except ANUSPLIN-VIC showing a SWE underestimation compared to observations. In the middle Fraser, the UW-VIC simulation is quite comparable to observations whereas the PCIC-VIC simulation underestimates SWE at Mission Ridge. Both ANUSPLIN-VIC and NARR-VIC underestimate SWE in the middle Fraser locations. The observed SWE values in the lower Fraser locations are not well captured by VIC, perhaps owing to the region's coastal influence and strong sensitivity to air temperatures (not shown). These results highlight the importance of accurate precipitation forcings to simulate SWE. Along with this, even small perturbations in air temperature can change the phase of precipitation, which directly contributes to changes in SWE accumulation.

### 3.2.2 Comparison of observed versus simulated runoff

The VIC simulated flows are routed to produce hydrographs for the Fraser River at Hope, BC (Fig. 6a). Comparison of simulated runoff with observations shows the highly consistent model performance for PCIC-VIC and UW-VIC simulations whereas the runoff is considerably lower for the ANUSPLIN-VIC simulation. The NARR-VIC hydrograph is comparable in magnitude with observations but the runoff timing is considerably shifted (~15 days) yielding more winter and spring runoff and earlier decline of flows in summer. The shift in the hydrograph is probably caused by the 2°C warmer air temperatures

causing earlier snowmelt. This finding was confirmed by a VIC sensitivity experiment where the air temperature was perturbed by 2°C while keeping the precipitation unchanged. Similar to the case of NARR-VIC results, the simulated SWE and runoff decreases with 2°C rises in air temperatures (Supplementary Fig. 5). The coefficient of variation in daily runoff for all four datasets reveals that variability in the PCIC-VIC and UW-VIC simulations is similar to observations (Fig. 6b). We further produced the hydrographs for the FRB's six major sub-basins to compare VIC simulation runs of each basin. Similar to the hydrograph of the Fraser River at Hope, the ANUSPLIN-VIC runoff shows considerable disagreement with the observed hydrograph, especially in the UF, QU and TN basin owing to the dry bias in its precipitation forcing. Moreover, NARR-VIC runoff is overestimated in the SU, NA and CH sub-basins whereas for UF, QU and TN, the simulated runoff underestimates observed flows. Consistent with spatial differences of mean air temperature and runoff (Supplementary Figs. 1 and 3), the warmer NARR air temperatures (compared to PCIC) over the SU, NA and CH sub-basins in winter and spring induce more snowmelt and hence overestimate runoff. In contrast, over the UF, QU and TN, the NARR air temperature is comparatively cooler in winter. This may reduce the snowmelt driven runoff causing underestimation over these sub-basins. The PCIC-VIC hydrographs are better in most of the basins with high NSE scores (Supplementary Table 2).

Differences seen in the FRB's flow magnitude and timing clarify the impact of forcing uncertainties on the simulations. Such variation in simulated runoff especially during the snow-melting period (Apr-Jul) is either due to the uncertain amount of precipitation or the magnitude of air temperature in the forcing datasets.

We further investigated differences in forcings and their VIC simulation based on their climatic trends. The monthly climate trends in air temperature, precipitation and simulated runoff (Supplementary Fig. 6) shows relatively similar warm air temperatures (up to 3°C in December) and the declined precipitation (mainly snowfall) during winter for all four forcing datasets. The magnitude of trends in the NARR dataset is somewhat lower for air temperature and higher for precipitation compared to the other three datasets. In the simulated runoff, the monthly variation of trends generally agrees among simulations, but the trends are weak in the ANUSPLIN-VIC and UW-VIC simulations whereas the PCIC-VIC and NARR-VIC simulations exhibit strong trends. In the NARR-VIC simulations, runoff trends are affected by lower air temperature and higher precipitation trends yielding increasing runoff. Grid-scale trends show widespread differences in the NARR-VIC runoff, particularly in the interior of the FRB when compared to ANUSPLIN-VIC, UW-VIC and PCIC-VIC monthly trends (Supplementary Fig. 7). All four simulations exhibit strong positive runoff trends in April followed by declining trends in May in the Rocky Mountains (the UF and TN sub-basins).

The inter-comparison analysis shows that the uncertainties in forcing datasets contribute substantially to the performance of the VIC model. This is consistent with studies reporting that the uncertainties in model structure contribute less to snowpack and runoff simulations (Troin et al., 2015, 2016), whereas the uncertainties in forcing datasets are the predominant sources of uncertainties (Kay et al., 2009; Chen et al., 2011). Using the NARR dataset, the systematic biases in simulations and the substantial effect of lateral boundary conditions on the performance of the regional model have also been identified in many other studies (de Elia et al., 2008; Eum et al., 2012).

While the small differences in precipitation are acceptable, the air temperature uncertainties play an especially important role in the hydrological simulations. In the FRB, air temperature controls summer water availability, making regional snowpacks more vulnerable to temperature-induced effects, rather than precipitation. Thus uncertainties in air temperatures are crucial for the runoff timing in hydrological simulations over the FRB rather than those in precipitation.

## 3.3 Uncertainty in Calibration Optimizer

We further investigated the uncertainty in the optimization of parameters during the calibration process. Many studies have evaluated the parameter uncertainties by adding random noise to the calibration parameters. We used a different approach by estimating the uncertainty in the MOCOM-UA optimizer used in the calibration of parameters. This was to estimate the optimizer uncertainty during the VIC calibration process using different values of initial parameters limits. The optimization process for the OPT1, OPT2, OPT3, OPT4 and OPT5 experiments required 39, 89, 61, 52, and 56 iterations, respectively, to optimize the b_inf, Ds, Ws, D2, D3 and Dsmax parameters to their final values (Table 3). The corresponding mean monthly (as the optimizer cannot utilize daily data) runoff for the Fraser River at Hope in the OPT1, OPT2, OPT3, OPT4 and OPT5 experiments are quite different when compared to observations (Fig. 8). The NSE scores reveal different accuracy for the five simulations even when the parameters' initial range in the OPT1, OPT3, OPT4 and OPT5 experiments is a subset of the OPT2 experiment. The optimization process for parameter calibration would require an expert's experience to set the initial parameter range to converge them to their optimal values. Note that if the initial parameter uncertainty distribution is set as wide as it is physically meaningful, then the

optimization will require more computational time to converge toward the Pareto optimum. However, to set the initial parameter limits, the subjective judgement and skill based on experience is needed.

While we performed many sets of experiments with different initial parameters, only OPT1's initial limits produced higher NSE and utilized less computational time. The estimation of hydrologic model parameters depends significantly on the availability and quality of the precipitation and observed streamflow data along with the accuracy of the routing model used. It is therefore important to consider bias correction of forcing datasets as part of automatic calibration. The observed streamflow data used to calibrate the model are often based on water levels that are converted to discharge by the use of a rating curve, which can also induce uncertainty in the observed discharge data. The overall conclusion of this analysis is that the automated optimizers used to converge calibration parameters still rely on the hydrologist's experience and some manual adjustment of initial calibration parameter ranges.

### 3.4 Uncertainty in calibration due to PDO Phases

The FRB streamflow varies from year to year as well on decadal timescales depending on the timing and magnitude of precipitation and air temperatures during the preceding winter and spring. Given that the FRB air temperature and precipitation are influenced by cool and warm phases of the PDO (Mantua et al., 1997; Fleming et al., 2010; Whitfield et al., 2010; Thorne and Woo, 2011), the choice of VIC calibration and validation periods may induce uncertainty in calibration. The influence of PDO phases in the forcing dataset can produce different snowpack and runoff levels in the hydrological simulation. The long term UW-VIC simulations (1949-2006) show higher mean SWE and runoff levels in a cool PDO phase (1949-1976) and lower mean values in a warm PDO phase (1979-2006) (Supplementary

Fig. 8). The interannual variations show earlier peak flows characterized by a warm PDO, in response to warmer basin conditions, increased rainfall, and earlier snowmelt. The VIC model calibrations may be biased towards hydrologic conditions of the warm and cold PDO phases and may induce uncertainties in the results. The model performance could be improved by calibrating and validating the model in the

same PDO phase (experiments PDO1, PDO2 and PDO5), i.e. the NSE coefficient is similar in the calibration and validation periods (Table 4). If the calibration is performed in the cool PDO phase and validation in the warm PDO phase (experiment PDO3), the NSE score decreases to 0.79 for the validation period since the model calibration is biased towards the cool conditions, simulating higher flows for the Fraser River at Hope owing to more snow and later snowmelt. The same is true if the

calibration and validation is performed in the warm and cool PDO phases, respectively (experiment PDO4). For each set of calibration experiments, the calibration parameters are different, which affects the formation of the snowpack, and the timing of snowmelt. Figure 9 shows observed and simulated runoff for the Fraser River at Hope revealing lower observed peak flows ~2.7 mm day$^{-1}$ in a warm PDO phase (PDO1) and higher peak flows ~3.3 mm day$^{-1}$ in a cool PDO phase (PDO2). Interestingly the UW

driven PDO simulations underestimate peak flows in the warm PDO phase and overestimate them in the cool PDO phase whereas the NSE coefficient for both the cool and warm PDO phases is almost equivalent (Table 4). The PDO4 and PDO5 experiments further support these findings.

This analysis reveals that the hydrological model performance changes considerably with different climatic conditions and the choice of the calibration and validation time periods, an important factor in

hydrological simulations. The proper implementation of a hydrological model requires a careful

calibration strategy to produce reliable hydrological information important for water resource management.

## 4. Conclusions

This study utilized ANUSPLIN, NARR, UW and PCIC observation-based gridded datasets to evaluate systematic inter-datasets uncertainties and their VIC simulated hydrological response over the FRB. The uncertainties involved in the optimization of model parameters and model calibration under cool and warm phases of the PDO were also examined.

The air temperature in the PCIC and UW datasets were comparable while the PCIC precipitation remains quite high in the Rocky Mountains compared to the UW and NARR datasets. The ANUSPLIN precipitation forcing had a considerable dry bias over mountainous regions of the FRB compared to the NARR, UW and PCIC datasets. The NARR winter air temperature was 2°C warmer than the other datasets over most of the FRB. The PCIC-VIC and UW-VIC simulations had higher NSE values and more reasonable hydrographs compared with observed flows for the Fraser River at Hope. Their performance for many of the FRB's major sub-basins remained satisfactory. The PCIC-VIC simulation revealed higher SWE compared to other datasets, probably due to its higher precipitation amounts. The ANUSPLIN-VIC simulation had considerably lower runoff and NSE values along with less SWE and snowmelt amounts owing to its reduced precipitation. The NARR dataset showed warm winter air temperatures, which influenced its hydrological response by simulating less SWE and decreased snowmelt, and hence lower runoff. The monthly trend analysis distinguished the NARR dataset by showing decreased trends in air temperature and increased trends in precipitation and its VIC driven

runoff. The elevation dependence of maxSWE showed disagreements over the higher elevations of the Rocky Mountains between simulations where the PCIC-VIC simulation overestimated SWE and ANUSPLIN-VIC resulted in underestimation. Furthermore the elevation dependent variation of the maxSWE-day fluctuated considerably between simulations.

The parametric uncertainty in the VIC calibration process revealed that the choice of the initial parameter range plays a crucial role in defining the model performance. During the PDO phases, choice of the calibration and validation time periods play a crucial role in defining the model hydrological response for the FRB. Model calibration was biased towards hydrologic conditions of the warm and cold PDO phases. The UW-VIC PDO simulations underestimated and overestimated the peak flows in

the warm and cool PDO phases, respectively.

This study's inter-comparison revealed spatial and temporal differences amongst the ANUSPLIN, NARR, UW and PCIC datasets over the FRB, which is essential to capture the uncertainties in modelling hydrologic responses. Overall, the PCIC and UW datasets had reliable results for the FRB snow hydrology whereas the ANUSPLIN and NARR datasets had issues with either precipitation or

with air temperature. The FRB snow-dominated hydrology and its complex elevation profile require highly accurate meteorological station densities to increase the reliability of the high resolution gridded datasets. While the air temperature plays a dominant role in the hydrological simulations, improving the quality of precipitation data can lead to more accurate hydrological responses in the FRB. Considerable precipitation bias can substantially degrade the model performance. There is the need for concrete

methods to deal with the increasing uncertainty associated with the models themselves, and with the observations required for driving and evaluating the models.

In this study, the FRB hydrological response varied considerably under different forcing datasets, modelling parameters and remote teleconnections. However there are other sources of uncertainties not discussed here that may establish a range of possible impacts on hydrological simulations. First, the hydrological model used in this study runs at a daily time step, which can be increased to hourly to refine the model performance. The lack of the representation of glaciers in the current version of the VIC model may induce uncertainties in model results. Along with these, the VIC simulations are also affected by intrinsic uncertainties in its parameterizations such as, for example, the representation of cold processes (e.g., snowpacks and soil freezing). The in situ soil moisture observations that are not necessarily representative of the model grid scale may also contribute to the overall uncertainties in the results. Finally, hydrological simulations are mainly validated using comparisons between simulated and observed flows, which depend on routing models that may contain structural uncertainties. Our future work will investigate such uncertainties using high temporal and spatial resolution hydrological models over the FRB.

**Acknowledgments**

This work was supported by the NSERC-funded Canadian Sea Ice and Snow Evolution (CanSISE) Network. This manuscript was motivated by the Eric Wood Symposium held at Princeton University on 2-3 June 2016. The authors are grateful to colleagues from the Pacific Climate Impacts Consortium (PCIC) for providing ongoing assistance with this research and to Dennis Lettenmaier at UCLA for providing assistance in the VIC model implementation. The authors are thankful to Michael Allchin (UNBC) for plotting Fig. 1a, Xiaogang Shi (Xi'an Jiaotong-Liverpool University) for development and

improvements of the UW dataset, and Do Hyuk Kang (NASA GSFC) for helping in setting up the VIC model over the FRB. Thanks to the anonymous referees and the handling editor for constructive comments that greatly improved the paper.

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

**Table 1: Description of VIC inter-comparison experiments performed using observational forcings.**

| VIC model Driving data | Time Span | Ensemble Runs | VIC configuration | Description |
|---|---|---|---|---|
| The Canadian Precipitation Analysis and the thin-plate smoothing splines (**ANUSPLIN**, Hopkinson et al., 2011) | 1979-1990 (Cal.) 1991-2006 (Val.) | 5 | Domain = 48°– 55°N and 119°– 131°W Resolution = 25 km × 25 km, Time Step = Daily Soil Layers = 3 Vertical elevation bands = 10 | Validation runs are initiated 5 times with different initial conditions. |
| North American Regional Reanalysis (**NARR**, Mesinger et al., 2006) | | | | |
| Pacific Climate Impacts Consortium (**PCIC**, Shrestha et al., 2012) | | | | |
| University of Washington (**UW**, Shi et al., 2013) | | | | |

**Table 2: Daily performance metrics for the VIC inter-comparison runs. Calibration (1979–1990) and validation (1991–2006) for Fraser River main stem at Hope, BC is evaluated using the Nash–Sutcliffe Efficiency (NSE) coefficient and correlation coefficient (r, all statistically-significant at $p < 0.05$).**

| Experiments | 1979-1990 Daily Calibration | | 1991-2006 Daily Validation | |
|---|---|---|---|---|
| | NSE | r | NSE | r |
| ANUSPLIN-VIC | 0.54 | 0.91 | 0.55 | 0.94 |
| NARR-VIC | 0.67 | 0.85 | 0.81 | 0.90 |
| PCIC-VIC | 0.90 | 0.96 | 0.90 | 0.95 |
| UW-VIC | 0.82 | 0.94 | 0.80 | 0.92 |

**Table 3: Parameters used to optimize during the calibration process for mean daily runoff for the Fraser River at Hope. OPT1, OPT2, OPT3, OPT4 and OPT5 are different experiments using same forcing data but with different initial range for each calibration parameter.**

| VIC model Calibration Parameters (units) | Description | Initial Range (Final Optimized Parameters) | | | | |
|---|---|---|---|---|---|---|
| | | Experiment OPT1 | Experiment OPT2 | Experiment OPT3 | Experiment OPT4 | Experiment OPT5 |
| b_inf | Controls the partitioning of precipitation (or snowmelt) into surface runoff or infiltration | 0.2-0.00001 (0.07) | 0.3-0.00001 (0.16) | 0.25-0. 10 (0.10) | 0.1-0.0001 (0.08) | 0.16-0.12 (0.12) |
| Ds | Fraction of maximum baseflow velocity | 0.1-0.000001 (0.05) | 0.9-0.00001 (0.09) | 0.30-0.04 (0.05) | 0.6-0.0001 (0.19) | 0.09-0.03 (0.05) |
| Ws | Fraction of maximum soil moisture content of the third soil layer at which nonlinear baseflow occurs | 0.6-0.20 (0.33) | 1.0-0.1 (0.49) | 0.65-0.20 (0.50) | 0.5-0.3 (0.42) | 0.35-0.20 (0.31) |
| D2 (m) | The second soil layer thicknesses, which affect the water available for transpiration | 1.0-0.7 (0.82) | 3.0-0.7 (1.02) | 0.80-0.70 (0.76) | 2.8-1.0 (1.07) | 0.80-0.70 (0.78) |
| D3 (m) | The third soil layer thicknesses, which affect the water available for baseflow | 2.5-0. 7 (1.66) | 5.5-0.7 (2.70) | 2.00-1.00 (1.82) | 3.0-1.0 (1.38) | 1.8-1.2 (1.76 ) |
| Dsmax (mm day-1) | Maximum baseflow velocity | 18.0-12.0 (16.0) | 30.0-12.0 (22.71) | 23.0-12.0 (14.28) | 18-12 (16.22) | 16-13 (14.11) |
| **Monthly NSE** | - | **0.93** | **0.84** | **0.92** | **0.89** | **0.91** |

**Table 4: Daily performance metrics for the UW forcing driven PDO runs. Calibration and validation for Fraser River main stem at Hope, BC is evaluated using the NSE coefficient using the dataset. See text for the detail of PDO experiments.**

| Experiment Name | Calibration | | Validation | |
|---|---|---|---|---|
| | NSE (Time period) | PDO Phase (Flows) | NSE (Time period) | PDO Phase (Flows) |
| **PDO1** | 0.84 (1981-1990) | Warm (low flows) | 0.84 (1991-2001) | Warm (low flows) |
| **PDO2** | 0.84 (1957-1966) | Cool (high flows) | 0.85 (1967-1976) | Cool (high flows) |
| **PDO3** | 0.84 (1967-1976) | Cool (high flows) | 0.79 (1977-1987) | Warm (low flows) |
| **PDO4** | 0.86 (1977-1987) | Warm (low flows) | 0.80 (1967-1976) | Cool (high flows) |
| **PDO5** | 0.89 (1991-2001) | Warm (low flows) | 0.87 (1981-1990) | Warm (low flows) |

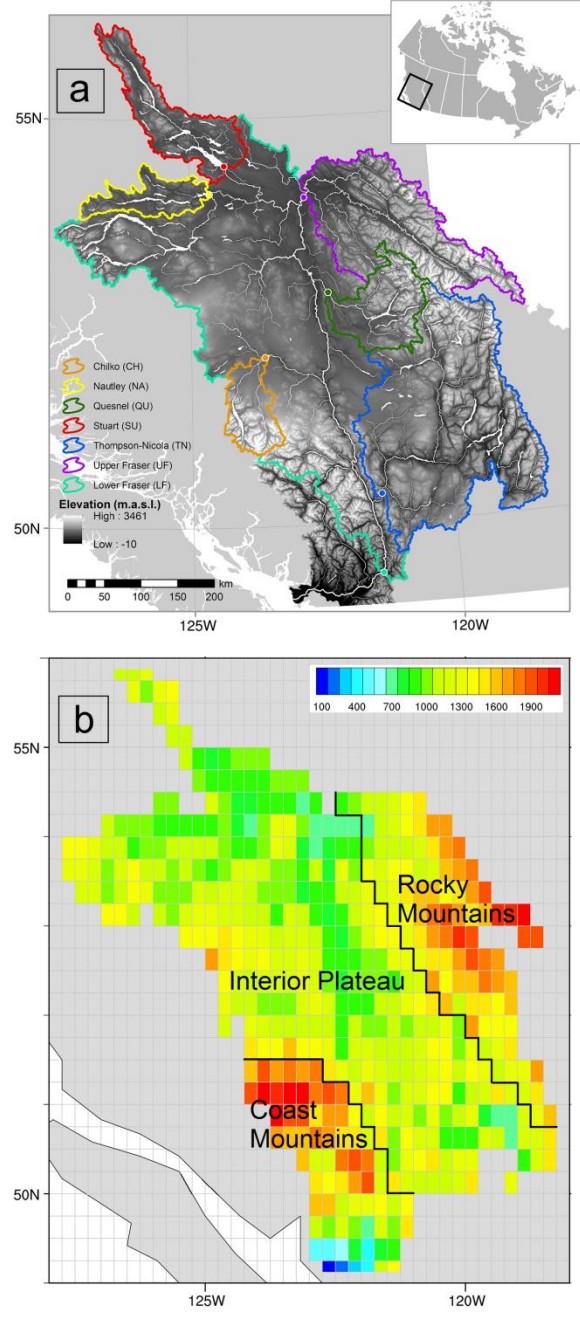

**Figure 1: (a) High-resolution digital elevation map of the FRB with identification of major sub-basins including the Fraser River main stem at Hope, BC. (b) FRB mean elevation (m) per VIC model gridcell. The location of the hydrometric gauge on the Fraser Rivers' main stem at Hope is highlighted in both plots with black circles.**

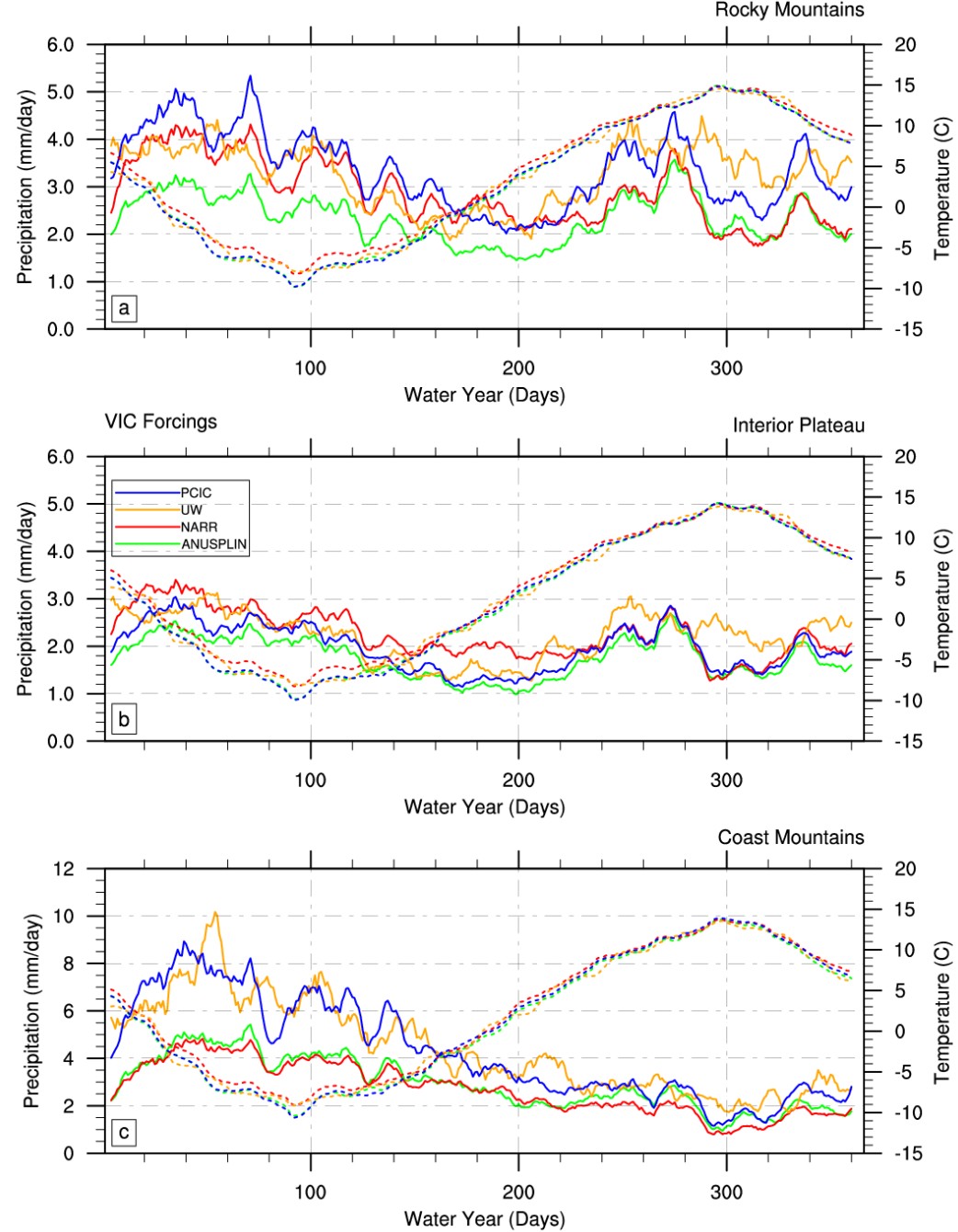

**Figure 2: Area-averaged time series of mean daily air temperature (dotted lines) and daily precipitation (solid lines) over the (a) Rocky Mountains, (b) Interior Plateau, and (c) Coast Mountains for the ANUSPLIN, NARR, UW and PCIC forcing datasets, water years 1979-2006. Water year starts on 1 October and ends on 30 September of the following calendar year.**

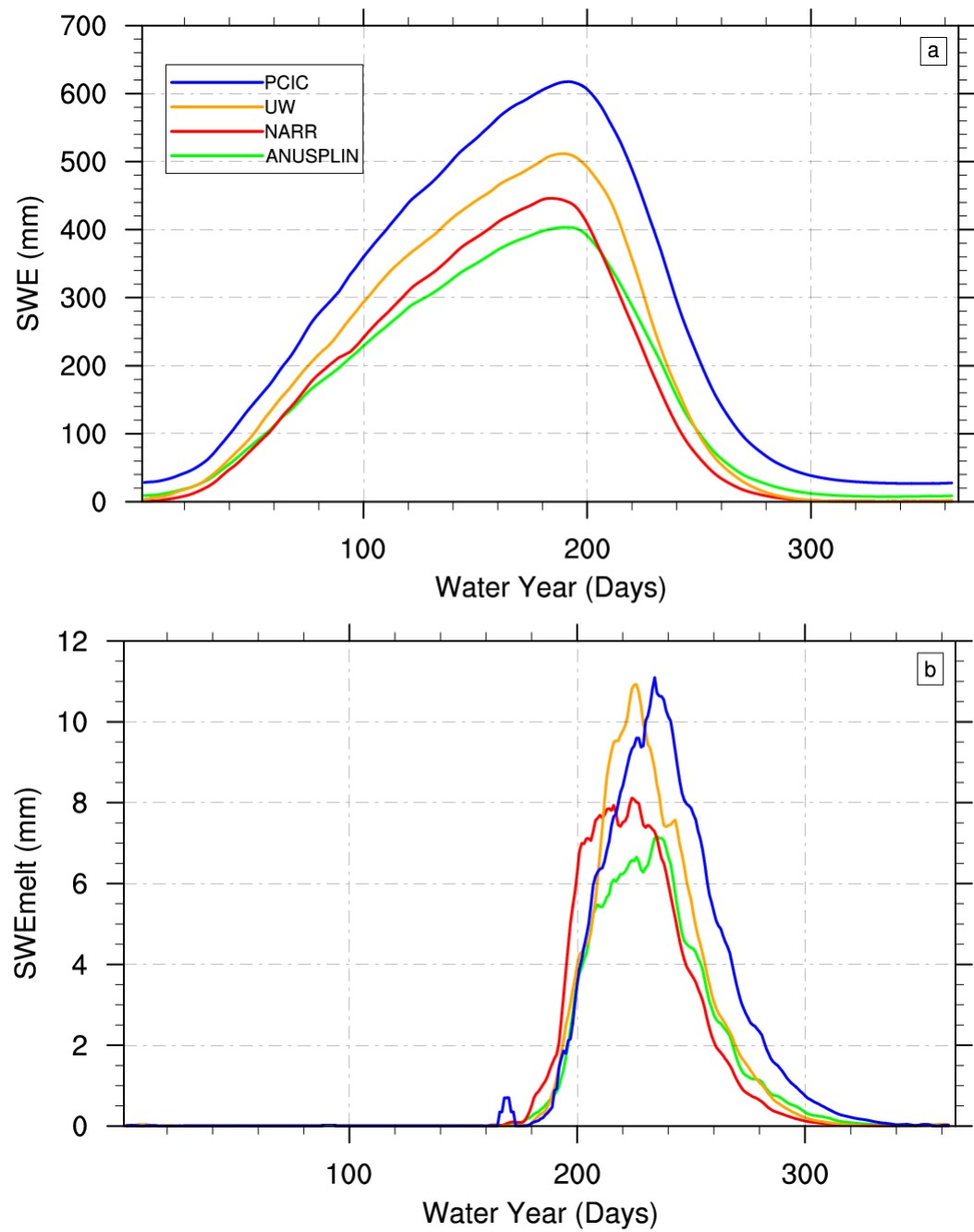

**Figure 3: Area-averaged time series of daily mean (a) SWE and (b) SWE$_{melt}$ for the ANUSPLIN-VIC, NARR-VIC, UW-VIC and PCIC-VIC simulations averaged over the Rocky Mountains, water years 1979-2006. Water year starts on 1 October and ends on 30 September of the following calendar year.**

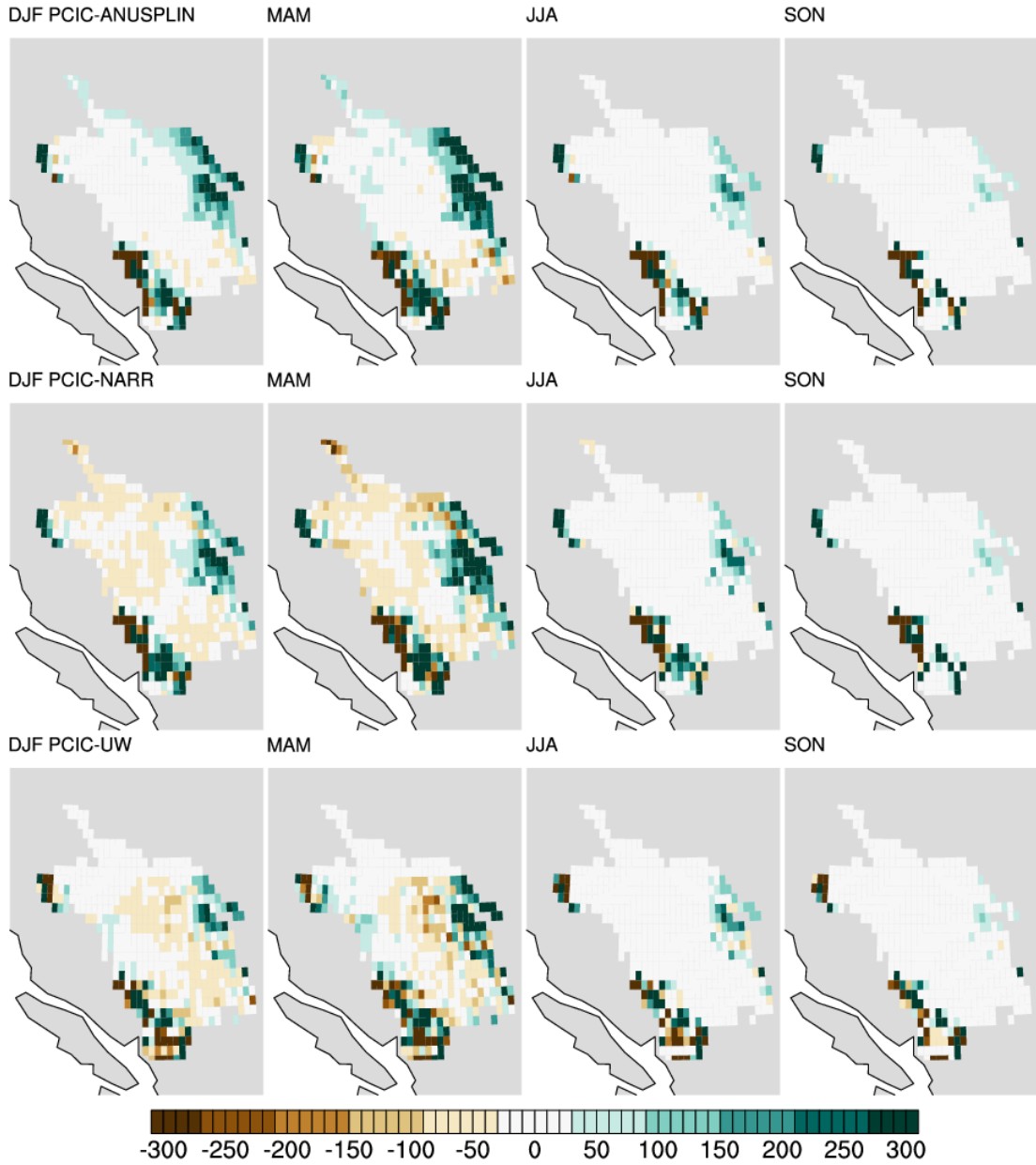

**Figure 4: Spatial differences of mean seasonal SWE (mm) based on PCIC-VIC minus ANUSPLIN-VIC (1st row), NARR -VIC (2nd row) and UW (3rd row) simulations, water years 1979-2006. DFJ, MAM, JJA and SON correspond to winter, spring, summer and autumn, respectively.**

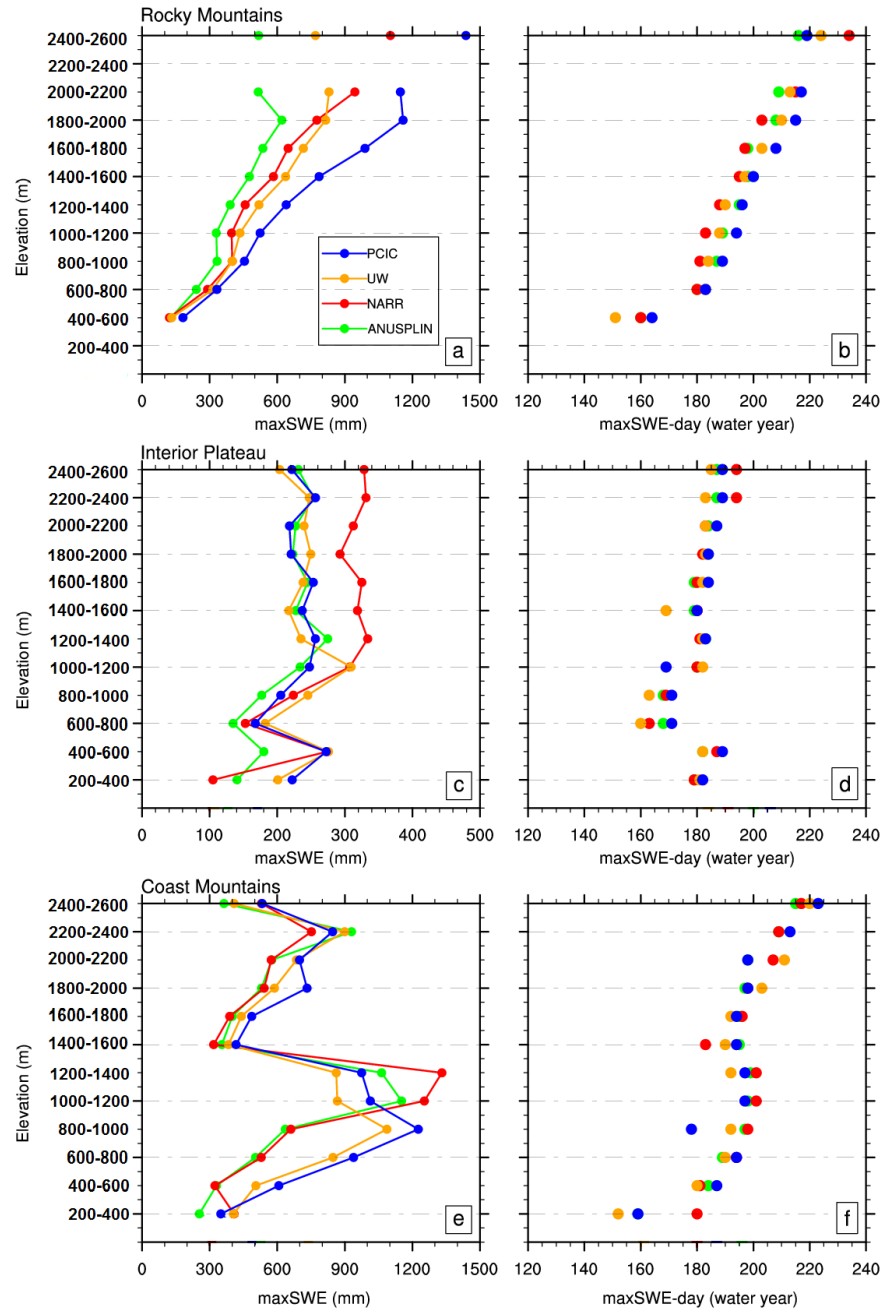

**Figure 5: Variation of (a, c, e) maxSWE and corresponding (b, d, e) maxSWE-day for the ANUSPLIN-VIC, NARR-VIC, UW-VIC and PCIC-VIC simulations averaged over the (a, b) Rocky Mountains, (c, d) Interior Plateau and (e, f) Coast Mountains, water years 1979-2006.**

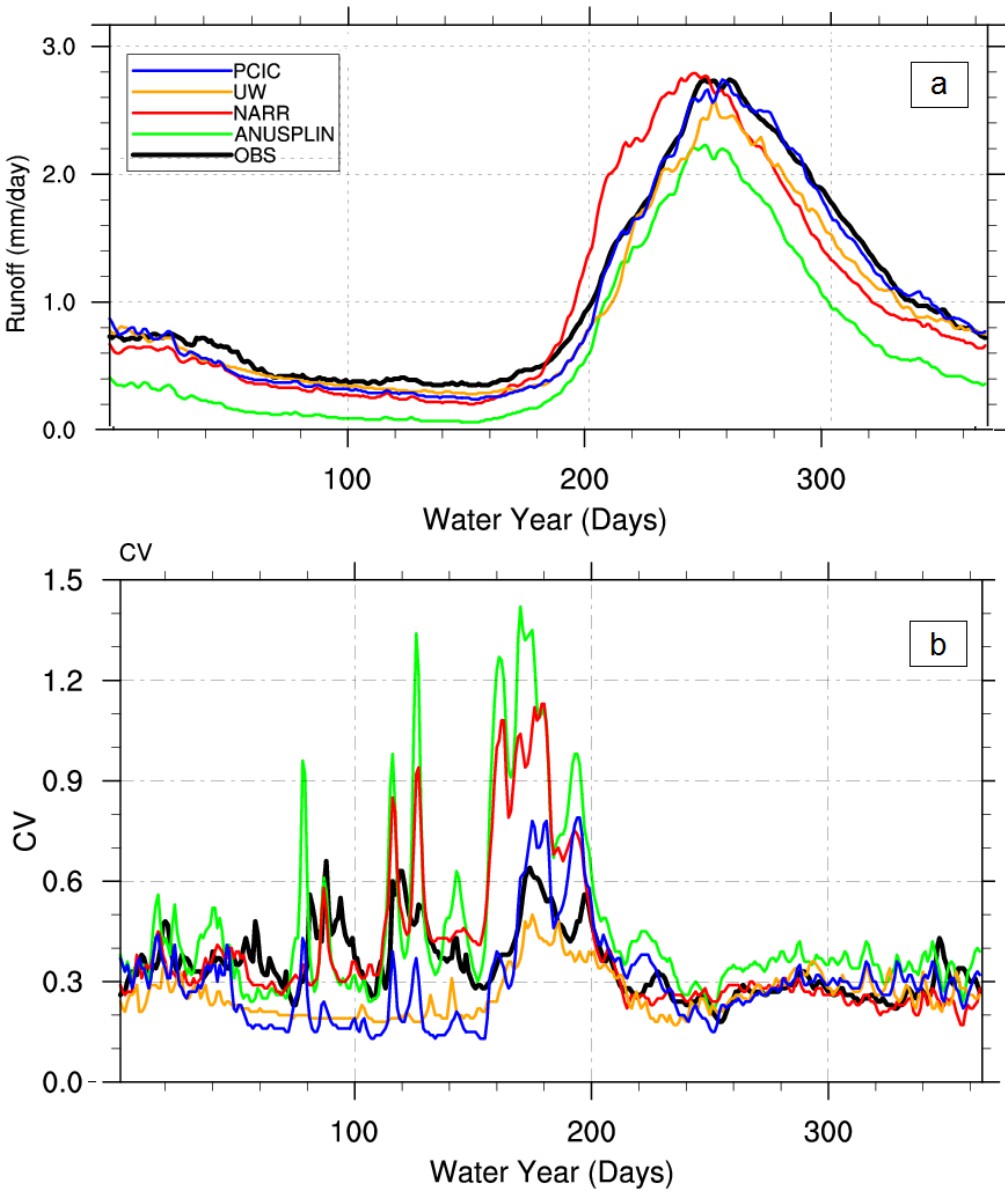

**Figure 6: The simulated and observed daily (a) runoff and (b) coefficient of variation (CV) for the Fraser River at Hope averaged over water years 1979-2006. An external routing model is used to calculate runoff for the ANUSPLIN-VIC, NARR-VIC, UW-VIC and PCIC-VIC simulations. Water year starts on 1 October and ends on 30 September of the following calendar year.**

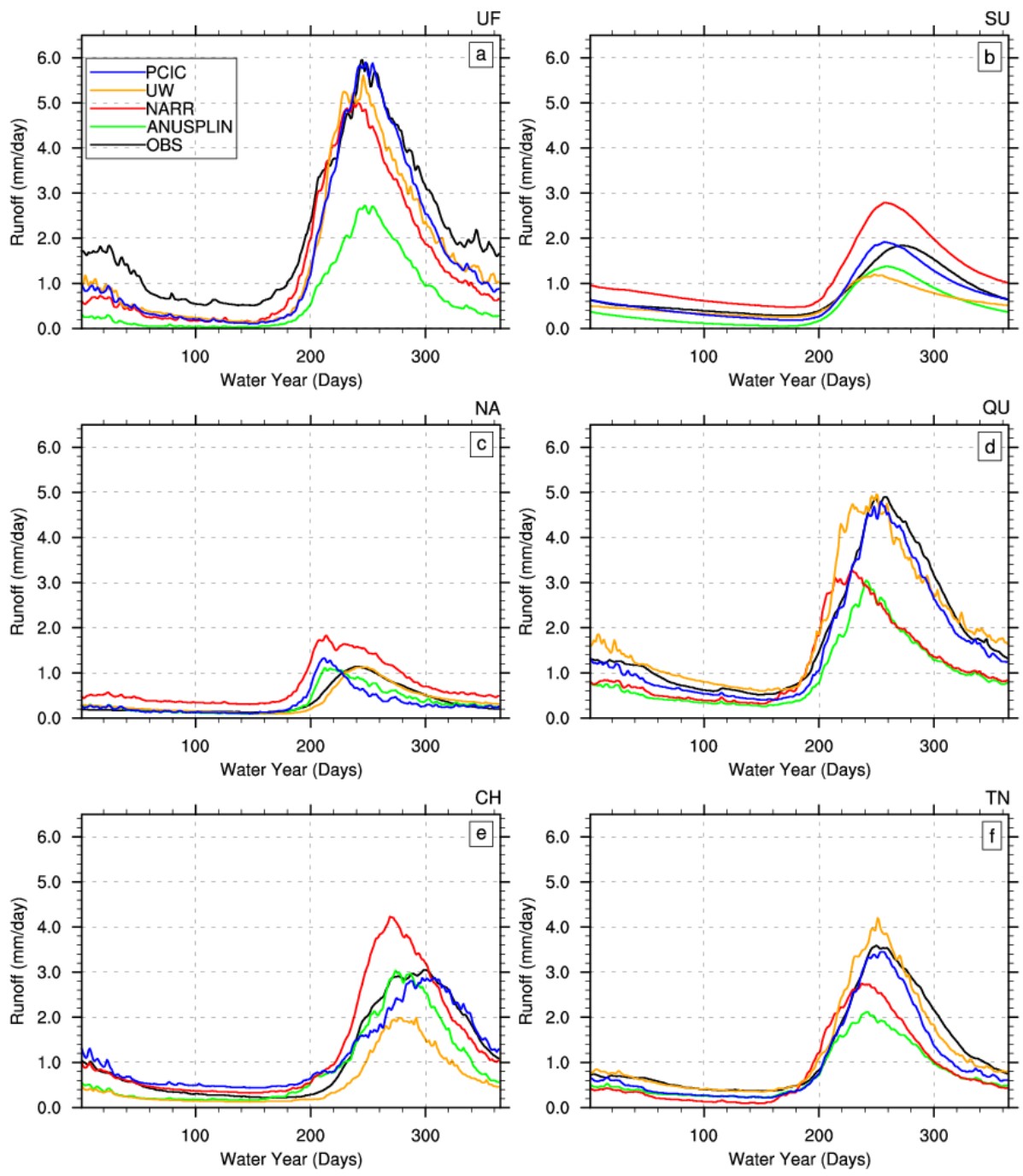

**Figure 7: Same as Figure 6a but for the FRB's six major sub-basins (a) Fraser-Shelley (UF), (b) Stuart (SU), (c) Nautley (NA), (d) Quesnel (QU), (e) Chilko (CH) and (f) Thompson-Nicola (TN).**

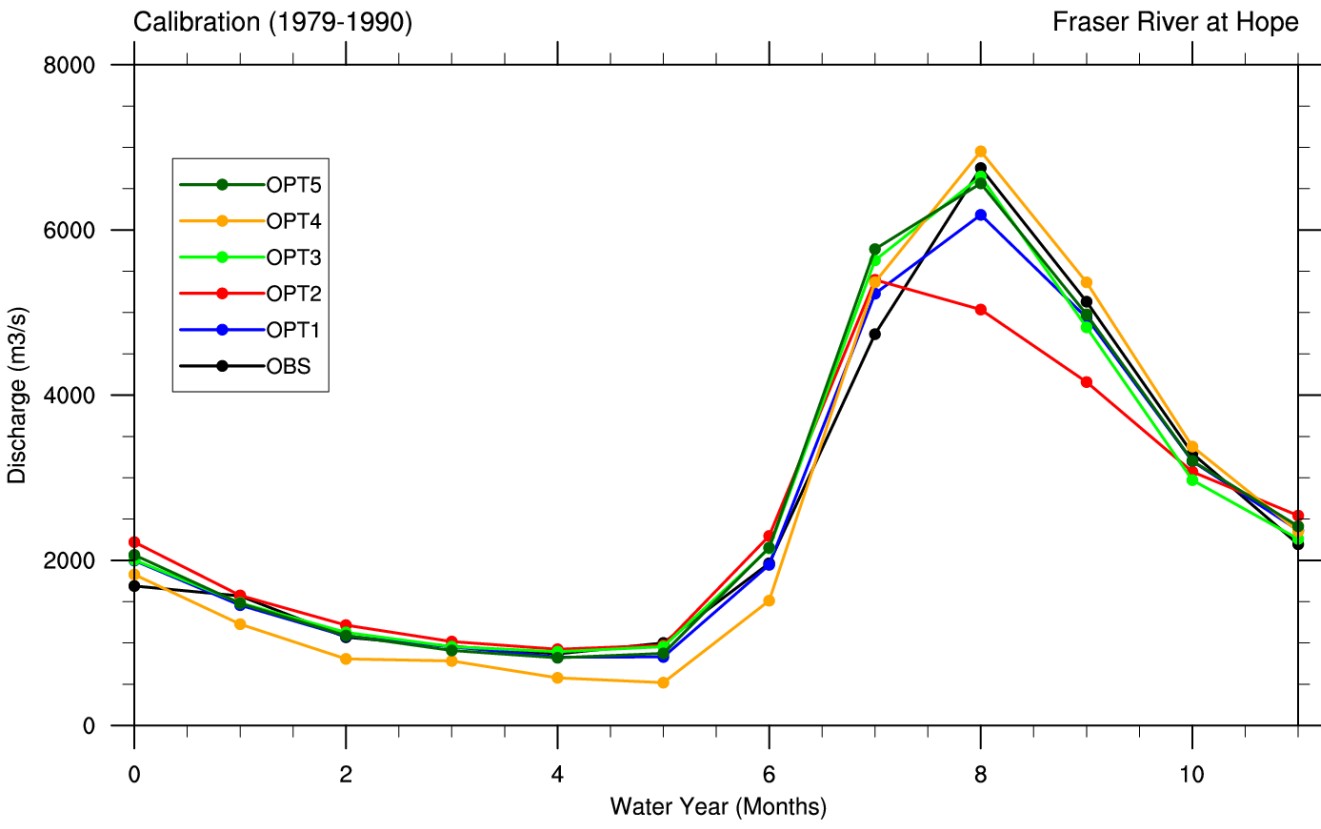

**Figure 8: UW-VIC simulations using five different parameter sets (labelled as OPT1, OPT2, OPT3, OPT4 and OPT5, see text and Table 3 for details) are compared for mean monthly discharge for the Fraser River at Hope during calibration period 1979-1990. The black curve represents observed monthly discharge.**

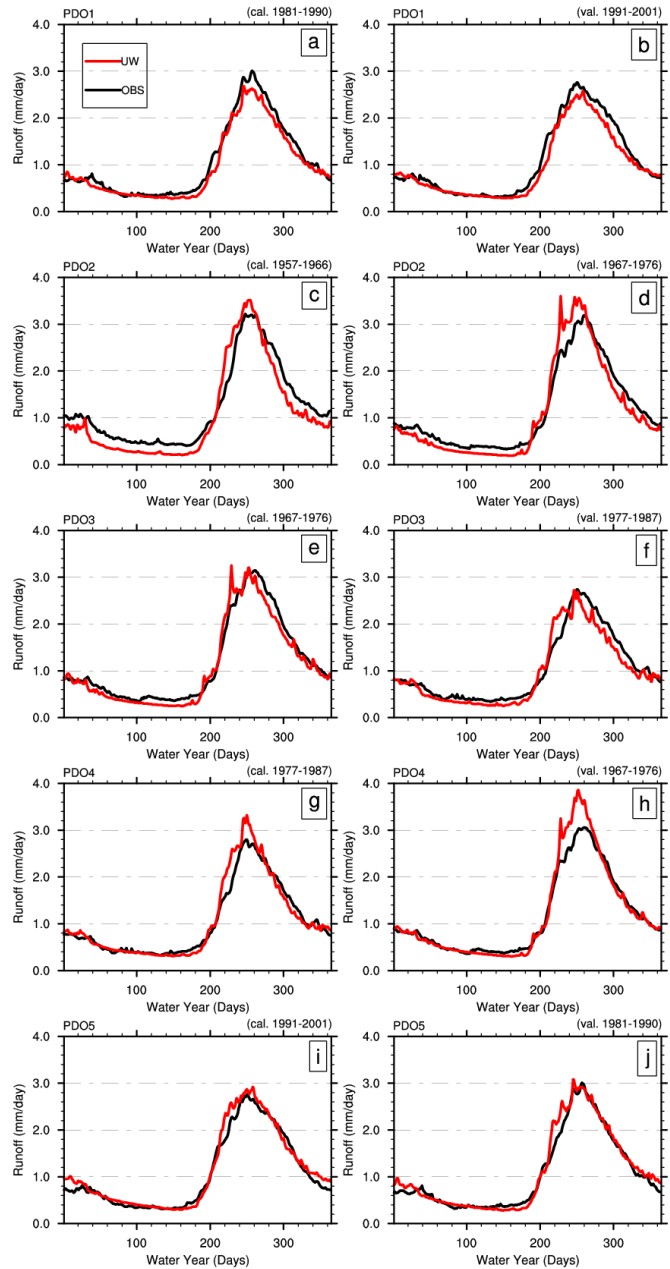

**Figure 9: UW-VIC simulated daily runoff during calibration (cal.) and validation (val.) for the Fraser River at Hope, BC.  PDO1, PDO2, PDO3, PDO4 and PDO5 refer to the VIC experiments performed during different experimental setups (see text and Table 4 for details). Water year starts on 1 October and ends on 30 September of the following calendar year.**