# Peer review of "Evaluating uncertainties in modelling the snow hydrology of the Fraser River Basin, British Columbia, Canada"

_Hydrology and Earth System Sciences, 2016_

## Referee Comment (RC1) · Anonymous Referee #1 · 19 Oct 2016

This manuscript provides an assessment of the uncertainties in hydrological modelling over the FRB, BC, Canada. The authors used four precipitation dataset and a calibration routine to quantify and characterize the uncertainties to related to meteorological and parameters estimates. The manuscript is well written and covers an interesting topic, however, I feel some issues have not been properly addressed. I suggest moderate revisions to the manuscript before publication. In general I think some of the procedures could be implemented more consistent to ensure that the conclusions of the paper are better

Major comments Page 11 Line 19-20 The authors want to study the impact of different forcing dataset and their related uncertainties in hydrological simulations. They

mention a couple of times that the errors in mountainous precipitation might cause significant discrepancies between dataset and therefore they use four different datasets to study that impact. Although the datasets all have their own resolution, the authors decide to bilinear interpolate all of them to a common grid of 25km without taking into account elevation corrections. I believe this procedure will add to the forcing uncertainty, especially for the coarser resolution since they are downscaled to a resolution without including the high resolution elevation data to correct for orographic effects. I think it would be good if the authors could provide some estimate for the uncertainty added by the bilinear interpolation without conditioning on the elevation profile.

Section 2.4.1 Why did the authors select these parameters? Is there sensitivity information that could be used to identify the most sensitive parameters? Maybe the impact of the routing model is more prominent, while it is not calibrated and doesn't account for the reservoirs and lakes present in the FRB. In addition, none of the snow parameters is calibrated, while the authors mention the importance of snow throughout the manuscript. Maybe a calibration on the snow processes (compaction, sublimation or just simple degree day factor), would further benefit the discharge simulation at the outlet and for the sub-catchments.

Page 15, why is on the PCIC forcing data used for the optimizer. This almost ensures that the PCIC will have the best performance in the following evaluation sections. It might be more interesting (and more work), to calibrate for every forcing dataset individual and then use these four parameter sets for the validation with every unique forcing dataset leading to sixteen combinations. This also gives you four simulations per forcing dataset as a result of the different calibrations. I know this is some work, but it is feasible. I feel it would lift the quality of the overall uncertainty analysis and thereby better support the conclusions of the paper.

Minor comments I have discussed the ANUSPLIN acronym with some colleagues over the lunch break. We believe the authors could maybe come up with a better name.

[Figure]

Page 3 Line 13-14: measurement of the response metric -> objective function and the calibration variable

Page 7 Line4: Why did the author select the PDO rather than the more influential ENSO signal for Western Canada

Page 11 Line 8The PGF is not really high temporal resolution, it uses satellite observations, which are corrected with gauges at monthly temporal resolution. I think it would be good to provide the reader with the data sources of the PGF. This is important to understand the performance of the WU dataset

Section 2.2 and 2.4, maybe combine these sections since they both cover VIC and could be easily combined into on VIC section. Otherwise move section 2.3 forward to have the two VIC sections following one another.

Page 12 Line 19, maybe remove the number of columns and rows, the domain would be sufficient

Page 13 Line 3 Citation, year could be without the brackets

Page 14 Line 18 Why not loop over the year 1979 rather than no spin-up. If the forcing of 1979 were to be recycled for five year and the stabilized ICs could then be used rather than no spin-up. This would ensure that the NARR simulation is more equal to the others and therefore the difference can be really attributed to the difference in forcing rather than a cold model start.

Page 14 Line 18-20 Once calibration.....(Table 1) -> please clarify. It is not entirely clear what you want to do here.

Page 18 Line 9-12 Do you have estimates of the cross-correlation between the precipitation products. Up to what extend are they derived from the same input data.

Page 22 Line 15 mentions a VIC sensitivity experiment, it would be great to show some of these results to get a better understanding of the model parameter uncertainty

Page 24 Line 13-14 "air temperatures are more crucial for hydrological simulations", I would argue that this is true for the timing, but not so much for the total streamflow volume (at least not for the FRB, where evap is low). Maybe rephrase to "are crucial for the runoff timing in hydrological simulations"

Figure 2 When does the water year start? October 1st? Please clarify the captions of the figures, they could be more self-explaining

Figure 7, What is a,b,c,d,e,f? No explanation, also not in the caption of Figure 6

[Figure]

---

## Referee Comment (RC2) · Anonymous Referee #2 · 1 Nov 2016

*Hydrol. Earth Syst. Sci. Discuss.*

Manuscript Number:  doi:10.5194/hess-2016-469, 2016

Title:  Evaluating uncertainties in modelling the snow hydrology of the Fraser River Basin, British Columbia, Canada

**General comments**

The objectives of this study, which focuses on the modelling of snow hydrology of the Fraser River Basin (FRB) of British Columbia (BC), Canada, using the Variable Infiltration Capacity (VIC) model forced with several high-resolution gridded climate datasets (*i.e.*, . ANUSPLIN, NARR, UW, and PCIC), were to comprehensively assess uncertainties related to:  (i) driving datasets, (ii) optimization of model parameters, and (iii) model calibration during cool and warm phases of the Pacific Decadal Oscillation (PDO).

This is a very well-written paper that deals with relevant subject matters to the hydrological modelling community.  The paper delivers sound results with respect to objective (i), however, I believe those associated with objectives (ii) and (iii) are somewhat incomplete.  All the results are discussed with respect to mean values.  I believe the strength of the paper would be increased if the interannual variability was discussed as well.  Moreover, at the end of the paper, the reader is left with an incomplete take-home message; which begs the following question: how could we use the outcomes of the paper if we were asked to run the model to answer questions related to hydroelectric development or other water resources management issue in the FRB?  Something is missing here, like a solid recommendation.  Here are a few suggestions I can take from the paper:  whenever hydrological modelling is performed one should: (i) analyse the stream flow record with respect to previously identified teleconnection correlations?  (ii) perform various calibrations in order to find the optimal set of parameter values? (iii) have several snow elevation bands, may be every 100 m?  The readers want a clear take-home message from the authors on how to use their findings. Incidentally, the authors made such a reference in the introduction of the paper, but did follow through in their conclusion (last sentence on p. 4, lines 18-20).

I have made suggestions in the following list of specific comments below on how to fulfill what I perceive as shortcomings.  As a side note, I found a bit difficult the exercise of jumping back and forth between the content of the main manuscript and that of the Supplemental File.  Perhaps, the introduction of the content of the Supplement File and the content of Section 3.1 could be transformed into a useful technical note.

That being said, I strongly encourage the authors to address these comments as I feel the paper could certainly be a significant contribution to the community.

**Specific comments**

- To my knowledge several researchers have conducted studies related to objective (i) of this paper and, thus, the authors should consult the following references and relate their work accordingly:
  - Essou, G.R.C., R. Arsenault, R., F.P. Brissette. 2016. Comparison of climate datasets for lumped hydrological modeling over the continental United States. *J. of Hydrology*, **537**: 334-345.
  - Essou, G.R.C., F. Saberly, P. Lucas-Picher, F.P. Brissette, A. Poulin. 2016. Can Precipitation and Temperature from Meteorological Reanalyses Be Used for Hydrological Modeling? *J. of Hydrometeorology*, DOI: 10.1175/JHM-D-15-0138.1
  - Arsenault, R., F.P. Brissette. 2014. Continuous streamflow prediction in ungauged basins: the effects of equifinality and parameter set selection on uncertainty in regionalization approaches. *Water Resour. Res.*, **50** (7): 6135–6153
  - Sabarly, F., G. Essou, P. Lucas-Picher, A. Poulin, F. Brissette, 2016: Use of Four Reanalysis Datasets to Assess the Terrestrial Branch of the Water Cycle over Quebec, Canada. *Journal of Hydrometeorology*, **17**, 1447-1466
- To my knowledge, two other studies have dealt with the problem of using calibration and validation series that have different climatological characteristics (objective (iii) of the paper), those of Klemeš (1986. Operational testing of hydrological simulation models. *Hydrolog. Sci. Journal* **31**(1): 13-24) and Seiller *et al*. (2012. Multimodel evaluation of twenty lumped hydrological models under contrasted climate conditions. *Hydrology and Earth System Sciences*, European Geosciences Union, **16** (4): 1171-1189.).
  - In the introductory section of the manuscript, I invite the authors to relate objective (iii) of the study with the content of the aforementioned papers.
- P. 5, line 19: the authors refer to Islam *et al*. (2016), but it is missing from the list of references.
- P. 14, lines 4-6, it is written: « The VIC model calibration is applied to the Fraser River's main stem at Hope, BC and the FRB's major sub-basins, namely the Upper Fraser at Shelley (UF), Stuart (SU), 5 Nautley (NA), Quesnel (QU), Chilko (CH) and Thompson-Nicola (TN) Rivers (Fig. 1a and Supplementary Table 1). »
  - Does this mean there is a set of parameter values different for each sub-basin and one for the drainage area between Hope and the outlet of each sub-basin? It is common practice to do so with many distributed models, but I do not see anywhere in the paper the performance of the model with respect to each sub-basin outlet. Have I missed something?
- P. 14, lines 16-18: to be consistent, should not the five-year spin up period been applied to all gridded datasets; that is from 1979 to 1985; or loop over at least a couple of years until convergence is achieved and then undertake the calibration exercise?
- P. 15, lines 4-11: Why using the PCIC gridded dataset for the optimizer uncertainty runs? Why modifying the parameter space? Why not conducting several optimizer

uncertainty runs and look at the various local optima?  That is what is usually done.  By reducing the parameter space, the authors are restricting, in theory, the number of local optima, yet there could still be several of them in this modified space.  At the end we get three sets of local optima, yet there are multiple other sets.  This is an interesting twist, but:

- o Please provide more details on this calibration strategy.  The outcome of this work is incomplete as I believe there should have been multiple calibrations for each experiment in order to fully illustrate the equifinality. Please discuss.
- P. 15, lines 12-21:  why not six (6) experiments to fully explore the question (2 cool phases x 3 warm phases)?
  - o This is an interesting experiment, yet I believe it is incomplete as not all possibilities were explored (see Seiller *et al.*, 2012).
  - o Are not there any other highly-impacting teleconnection that affect the basin hydrology (*e.g.*, ENSO)?
- P. 16, lines 2-3:  Perhaps the three regions mentioned here should be depicted on the related Figures, that those figures where knowledge of the region is warranted to follow what is presented and discussed in the paper (Figure 4 in the manuscript and Figures 1-3 and 6 in the Supplement File).
- Given the results introduced in Section 3.1, I think it would have been interesting to have the following calibration parameters:  snow-to-rainfall temperature threshold and a vertical temperature gradient.
  - o Does the VIC model have such parameters? If that is the case, why not calibrating them?
  - o Is there a publication that has introduced a formal sensitivity analysis of the VIC model?
  - o Why focusing on soil-related parameters solely and not others as well? In other words, the authors should first introduce to the readers all the model parameters and provide a rationale for their choice.
- P. 19, lines 12-15: « The lower SWE in the NARR-VIC simulation is probably due to the warmer air temperature during winter and spring (Fig. 2b). Winter temperatures being warmer in the NARR dataset may alter the phase of precipitation partitioning with more rainfall than snowfall, and hence less SWE in the NARR-VIC simulation. »
  - o This is formulated as a hypothesis, but do not the authors have the input precipitation dataset to validate this hypothesis?
- P.20, line 4: what about a sensitivity analysis of the impact of the number of elevation bands on the output variables of interest? Why ten in other words, why not twenty?
- P.22, Section 3.2.2:  for each gridded dataset – VIC simulation what about calculating a runoff to precipitation coefficient to make inferences about model robustness?
- P.24, line 5: « …(Troin *et al.*, 2015, 2016)…» Please check as there is only one Troin *et al.* in the list of reference!

- P.26, lines 17-19: « For each set of calibration experiments, the calibration parameters are different, which affects the air temperature lapse rates and thus evapotranspiration, the formation of the snowpack, and the timing of snowmelt. »
    - I thought the temperature lapse rate was not a calibration parameter or is it?
- P.28, lines 5-6: « The trend analysis further highlighted uncertainties of the NARR Dataset. »
    - Uncertainties in terms of what...I am not sure I follow well the authors here. Please provide additional details.
- P29, line 3-4: is not the following sentence « Improving the quality of precipitation data may thus lead to more accurate hydrological responses in the FRB. » contradicting this other sentence « Thus uncertainties in air temperatures are more crucial for hydrological simulations over the FRB rather than those in precipitation» on page , lines 13-14?
    - Please help me here!
- P.29, line 12: not being familiar with the FRB, I was a bit surprise to learn here that the authors had never mentioned in the description of the basin that there are glaciers in the study area!
- P.29, lines 14-15: « Such structural uncertainties need to be evaluated by examining multiple hydrological models over the same region. »
    - That is fine to mention that, but this statement does not have anything to do with what was presented in the paper...why bring this issue and others up at this point, on top of it in the conclusion...maybe there should be a subsection untitled recommendations for future work

**Figures**

- Figures 2, 3, 6 & 7
    - What about producing a graph illustrating the daily values of the coefficient of variation so the readers can appreciate the interannual variablity for each data set?
- Figure 2
    - Days with January 1st as Julian day 1? In other words, what are the actual Julian days defining the water Year (1 October to 30 September of the following calendar year)? This comment apply to all relevant graphs introduced in the paper and supplement file.
- Figure 5
    - Boxplots would have been useful to highlight the interannual variability.

**Other comments**

Other specific comments are all related to verb tense can be found in the annotated manuscript. That is, to improve the reading and understanding I suggest using the past tense instead of the

present tense.  This may be viewed as an old fashion way of reporting scientific work. Now, I am also open to a strong argument to refute my point of view.

**Answer to traditional questions**

Is the paper free of errors in logic?

Yes

Do the conclusions follow from the evidence?

Yes.

Are alternative explanations explored as appropriate?

Yes.

Are biases, limitations, and assumptions clearly stated, and uncertainty quantified?

Yes.

Is methodology explained in sufficient detail so that the paper's scientific conclusions could be tested by others?

No, see the above list of specific comments.

Is previous work and current understanding cited and represented correctly?

No, see the above list of specific comments.

Is information conveyed clearly enough to be understood by the typical reader?

Yes

Are all figures and tables necessary, appropriate, legible, and annotated (as appropriate)?

Yes

[revised manuscript text omitted]

**Page : 12**

Nombre : 1     Auteur :   Sujet : Texte inséré  Date : 2016-10-29 10:11:10
d

Nombre : 2     Auteur :   Sujet : Texte inséré  Date : 2016-10-29 10:11:36
d

Nombre : 3     Auteur :   Sujet : Texte inséré  Date : 2016-10-29 10:11:57
d

Nombre : 4     Auteur :   Sujet : Texte inséré  Date : 2016-10-29 10:14:10
d

[Figure]

vertical snow elevation bands. Once an individual VIC simulation is completed, the runoff for the basin

is extracted at an outlet point of the given sub-basin, using an external routing model that simulated a

channel network (adapted from Wu et al. (2011)) with several nodes (Lohmann et al., 1996; 1998a, b).

Streamflow is converted to areal runoff by dividing it by the corresponding sub-basin area. Daily runoff

5  at the outlet cell is integrated over time to obtain total annual runoff for a selected basin. Other than the

calibration parameters, the soil and vegetation parameters, leaf area index (LAI) and albedo data are

kept identical as per the VIC model implementation to the FRB reported in Kang et al. (2014).

**2.4.1 Calibration**

To explore the feasible parameter space, we use the University of Arizona multi-objective complex

10  evolution (MOCOM-UA) optimizer for the VIC calibration process (Yapo et al., 1998; Shi et al., 2008).

MOCOM-UA searches a set of VIC input parameters using the population method to maximize the

Nash–Sutcliffe efficiency (NSE) coefficient (Nash and Sutcliffe, 1970) between observed and simulated

runoff. Six soil parameters are used in the optimization process, i.e. b_infilt (a parameter of the variable

infiltration curve), Dsmax (the maximum velocity of base flow for each gridcell), Ws (the fraction of

15  maximum soil moisture where nonlinear base flow occurs), D2 and D3 (the depths of the second and

third soil layers), and Ds (the fraction of the Dsmax parameter at which nonlinear base flow occurs).

Final values of these six calibrated parameters are estimated for each forcing dataset by a number of

simulation iterations minimizing the difference between the simulated and observed monthly flow.

[Figure]

While the MOCOM-UA automated optimization process utilizes monthly streamflow in calibration, we evaluated the overall model performance on daily time scales using NSE and correlation performance metrics.

The VIC model calibration is applied to the Fraser River's main stem at Hope, BC and the FRB's major
5   sub-basins, namely the Upper Fraser at Shelley (UF), Stuart (SU), Nautley (NA), Quesnel (QU), Chilko (CH) and Thompson-Nicola (TN) Rivers (Fig. 1a and Supplementary Table 1). These sub-basins contribute 75% of the annual observed Fraser River discharge at Hope, BC with the largest contributions from the TN, UF and QU sub-basins (Déry et al., 2012).

**2.4.2 Experiments**

10   A series of different VIC experiments are performed to i) compare the VIC model's response when driven by different forcings, ii) evaluate the uncertainties related to the VIC optimizer, and iii) investigate the effect of PDO teleconnections on the VIC calibration and validation time periods. These experiments are categorized as below:

1) **Inter-comparison runs:** The VIC model is driven by each forcing dataset for 28 years (1979 to
15       2006) with 1979-1990 as the calibration period and 1991-2006 as the validation period using the MOCOM-UA optimizer. The VIC simulations driven by ANUSPLIN, UW and PCIC forcings are initiated five years prior to the year 1979 to allow model spin-up time. Since NARR is not available until 1979, the model spin up time is not considered. Once calibration is completed, validation runs are initiated with five different initial conditions (lagged by 24-hour intervals) to avoid any random
20      uncertainties such as those in initial conditions (Table 1). The ANUSPLIN, NARR, UW and PCIC

**Page : 14**

| | | | |
|---|---|---|---|
| Nombre : 1 | Auteur : | Sujet : Texte inséré | Date : 2016-10-29 10:18:23 |

were

| | | | |
|---|---|---|---|
| Nombre : 2 | Auteur : | Sujet : Texte inséré | Date : 2016-10-29 10:18:32 |

(

| | | | |
|---|---|---|---|
| Nombre : 3 | Auteur : | Sujet : Texte inséré | Date : 2016-10-29 10:18:43 |

(

| | | | |
|---|---|---|---|
| Nombre : 4 | Auteur : | Sujet : Texte inséré | Date : 2016-10-29 10:18:59 |

(

| | | | |
|---|---|---|---|
| Nombre : 5 | Auteur : | Sujet : Texte inséré | Date : 2016-10-29 10:19:34 |

were

| | | | |
|---|---|---|---|
| Nombre : 6 | Auteur : | Sujet : Texte inséré | Date : 2016-10-29 10:19:28 |

follows

| | | | |
|---|---|---|---|
| Nombre : 7 | Auteur : | Sujet : Texte inséré | Date : 2016-10-29 10:19:48 |

was

| | | | |
|---|---|---|---|
| Nombre : 8 | Auteur : | Sujet : Texte inséré | Date : 2016-10-29 10:23:45 |

were

[revised manuscript text omitted]

**Page : 16**

[revised manuscript text omitted]

**Page : 20**

| | | | | |
|---|---|---|---|---|
| Nombre : 1 | Auteur : | Sujet : Texte inséré | Date : 2016-10-29 11:16:44 | |
| was | | | | |

| | | | | |
|---|---|---|---|---|
| Nombre : 2 | Auteur : | Sujet : Texte inséré | Date : 2016-10-29 11:16:33 | |
| d | | | | |

| | | | | |
|---|---|---|---|---|
| Nombre : 3 | Auteur : | Sujet : Texte inséré | Date : 2016-10-29 11:16:53 | |
| ed | | | | |

| | | | | |
|---|---|---|---|---|
| Nombre : 4 | Auteur : | Sujet : Texte inséré | Date : 2016-10-29 11:17:01 | |
| ed | | | | |

| | | | | |
|---|---|---|---|---|
| Nombre : 5 | Auteur : | Sujet : Texte inséré | Date : 2016-10-29 11:17:11 | |
| d | | | | |

snowpacks. For the Interior Plateau and the Coast Mountains, no consistent pattern of maxSWE-day variation exists for any particular simulation.

**3.2.1 Simulated SWE comparison with observations**

As mentioned earlier, all gridded climate forcing datasets are based on station observations. The density

5 of stations in the FRB's mountainous regions remains quite low and therefore induces higher uncertainties in the observational gridded products. It is important to quantify the spatial discrepancy between the simulated (0.25° gridcell) and observed (snow pillow station dataset) SWE that may lead to an uncertainty in snow estimations by models (Elder et al., 1991; Tong et al., 2010). We use
observed SWE from BC snow pillow sites and the VIC simulated SWE data over the same elevation and

10 overlapping continuous time periods at four different locations in the upper and middle Fraser where a high volume of SWE accumulates seasonally.

The daily time series of VIC simulated SWE (Supplementary Fig. 4) follows the observed interannual variability in snow accumulation but with considerable differences across simulations. The PCIC-VIC simulation accumulates too much SWE compared to observations in the gridcell corresponding to the

15 Yellowhead location. This overestimation is further explored for Yellowhead by expanding the time series back to 1979 (not shown), which reveals issues with PCIC precipitation data only during 1996-2004 exhibiting considerable above normal anomalies at Yellowhead. While ANUSPLIN-VIC shows lower SWE amounts, the NARR-VIC and UW-VIC simulations reproduce the observed variation quite reasonably for Yellowhead. For McBride, all simulations are more or less comparable except

20 ANUSPLIN-VIC showing a SWE underestimation compared to observations. In the middle Fraser, the

**Page : 21**

Nombre : 1   Auteur : Sujet : Texte inséré Date : 2016-10-29 11:22:25

d

[Figure]

UW-VIC simulation is quite comparable to observations whereas the PCIC-VIC simulation underestimates SWE at Mission Ridge. Both ANUSPLIN-VIC and NARR-VIC underestimate SWE in the middle Fraser locations. The observed SWE values in the lower Fraser locations are not well captured by VIC, perhaps owing to the region's coastal influence and strong sensitivity to air

5    temperatures (not shown). These results highlight the importance of accurate precipitation forcings to simulate SWE. Along with this, even small perturbations in air temperature can change the phase of precipitation, which directly contributes to changes in SWE accumulation.

**3.2.2 Simulated Runoff comparison with observations**

The VIC simulated flows are routed to produce hydrographs for the Fraser River at Hope, BC (Fig. 6).

10   Comparison of simulated runoff with observations shows the highly consistent model performance for PCIC-VIC and UW-VIC simulations whereas the runoff is considerably lower for the ANUSPLIN-VIC simulation. The NARR-VIC hydrograph is comparable in magnitude with observations but the runoff timing is considerably shifted (~15 days) yielding more winter and spring runoff and earlier decline of flows in summer. The shift in the hydrograph is probably caused by the 2°C warmer air temperatures

15   causing earlier snowmelt. This finding is
 confirmed by a VIC sensitivity experiment where the air temperature is perturbed by 2°C while keeping the precipitation unchanged. Similar to the case of NARR-VIC results, the simulated SWE, snowmelt and runoff decreases with 2°C rises in air temperature forcings. We further produce the hydrographs for the FRB's six major sub-basins to compare VIC simulation runs of each basin. Similar to the hydrograph of the Fraser River at Hope, the

20   ANUSPLIN-VIC runoff shows considerable disagreement with the observed hydrograph, especially in the UF, QU and TN basin owing to the dry bias in its precipitation forcing. Moreover, NARR-VIC

**Page : 22**

| | | | |
|---|---|---|---|
| Nombre : 1 | Auteur : | Sujet : Texte inséré | Date : 2016-10-29 11:31:40 |

was

| | | | |
|---|---|---|---|
| Nombre : 2 | Auteur : | Sujet : Texte inséré | Date : 2016-10-29 11:32:11 |

was

| | | | |
|---|---|---|---|
| Nombre : 3 | Auteur : | Sujet : Texte inséré | Date : 2016-10-29 11:33:27 |

d

[revised manuscript text omitted]

---

## Author Comment (AC1) · 8 Dec 2016

Thank you for the opportunity to submit our response to referees' comments on our manuscript titled "Evaluating uncertainties in modelling the snow hydrology of the Fraser River Basin, British Columbia, Canada" by Siraj Ul Islam and Stephen J. Déry (Reference # HESS-2016-469). We are thankful to anonymous Referees #1 and #2 for their comprehensive and constructive comments on our manuscript. We fully recognize and appreciate the referees' reports on our research. Indeed, incorporating their insights will lead us to improve our manuscript through the revision process. We are thus taking full consideration of the referees' comments and are preparing detailed responses listing a point-by-point response to each comment/suggestion along with the information on how the paper is being revised as per the two anonymous referees' suggestions. A complete and detailed response document will be submitted once a decision has been made on our discussion paper. Here we provide a general overview of our responses (in bold italic) to the major comments submitted by each referee. All minor comments will be addressed in our detailed response document.

Please let us know if additional information or points of clarification are required.

**Referee # 1:**
This manuscript provides an assessment of the uncertainties in hydrological modelling over the FRB, BC, Canada. The authors used four precipitation dataset and a calibration routine to quantify and characterize the uncertainties to related to meteorological and parameters estimates. The manuscript is well written and covers an interesting topic, however, I feel some issues have not been properly addressed. I suggest moderate revisions to the manuscript before publication. In general I think some of the procedures could be implemented more consistent to ensure that the conclusions of the paper are better.

***We recognize that there are complexities in our procedures that were not thoroughly addressed. In our revision process, we are paying special attention to ensure our results and discussion is more consistent by means of additional hydrological model simulations and analyses.***

Page 11 Line 19-20 The authors want to study the impact of different forcing dataset and their related uncertainties in hydrological simulations. They mention a couple of times that the errors in mountainous precipitation might cause significant discrepancies between dataset and therefore they use four different datasets to study that impact. Although the datasets all have their own resolution, the authors decide to bilinear interpolate all of them to a common grid of 25km without taking into account elevation corrections. I believe this procedure will add to the forcing uncertainty, especially for the coarser resolution since they are downscaled to a resolution without including the high resolution elevation data to correct for orographic effects. I think it would be good if the authors could provide some estimate for the uncertainty added by the bilinear interpolation without conditioning on the elevation profile.

***We agree that the elevation correction should be explicitly accounted for in the interpolation methodology to correct for orographic effects. Such correction is highly important when interpolating from coarser to higher spatial resolutions, especially in mountainous terrain (Dodson and Marks, 1997). However, in our interpolation methodology the NARR (32 km) dataset is interpolated from coarse resolution curvilinear grids to slightly higher (25 km) resolution rectilinear grids. On the other hand, both PCIC (6 km) and ANSUPLIN (10 km)***

*datasets are interpolated to a coarser resolution (25 km). Interpolating NARR datasets from 32 km to 25 km resolution does not induce much elevation dependent uncertainties since the change in orography remains minimal between mean elevations at 25 km and 32 km grid resolutions (Figure A). Thus the relationship of atmospheric variables such as air temperature with elevation remains nearly identical at both resolutions. In the revision, we will nonetheless highlight this source of uncertainty in our methodology section and will add details about the bilinear technique used in the interpolation process.*

[Figure]

*Figure A: Comparison of mean elevation (m) at 25 km and 32 km resolutions over the part of British Columbia covering Fraser River Basin.*

Section 2.4.1 Why did the authors select these parameters? Is there sensitivity information that could be used to identify the most sensitive parameters? Maybe the impact of the routing model is more prominent, while it is not calibrated and doesn't account for the reservoirs and lakes present in the FRB. In addition, none of the snow parameters is calibrated, while the authors mention the importance of snow throughout the manuscript. Maybe a calibration on the snow processes (compaction, sublimation or just simple degree day factor), would further benefit the discharge simulation at the outlet and for the sub-catchments.

*We selected these parameters for calibration based on the manual calibration experience from previous studies such as by Nijssen et al. (1997), Su et al. (2005), Shi et al. (2008), Kang et al. (2015, 2016) and Islam et al. (2016). VIC is a physically based hydrologic model that has many (about 20, depending on how the term "parameter" is defined) parameters that must be specified. However, the usual implementation approach involves the calibration of only six soil parameters. Such parameters have the largest effects on the hydrograph shape and are the most sensitive parameters in the water balance components (Nijssen et al. 1997; Su et al. 2005). These parameters must be estimated from observations, via a trial and error procedure that leads to an acceptable match of simulated discharge with observations.*

*For the snow calibration, we have fixed the value of thresholds for maximum (at which snow can fall) and minimum (at which rain can fall) air temperature as 0.5°C and -0.5°C, respectively. These values were adjusted based on the region's climatology and are kept constant for all the simulations in the global control file. Parameters related to the snow albedo are adjusted using the traditional VIC algorithm based on the US Army Corps of Engineers empirical snow albedo decay curves for transitions from snow accumulation to ablation. We will be adding this detail in Section 2.4.1 in our revision process.*

Page 15, why is on the PCIC forcing data used for the optimizer. This almost ensures that the PCIC will have the best performance in the following evaluation sections. It might be more interesting (and more work), to calibrate for every forcing dataset individual and then use these four parameter sets for the validation with every unique forcing dataset leading to sixteen combinations. This also gives you four simulations per forcing dataset as a result of the different calibrations. I know this is some work, but it is feasible. I feel it would lift the quality of the overall uncertainty analysis and thereby better support the conclusions of the paper.

*As discussed in our manuscript, the VIC model is quite sensitive to the meteorological forcing data, most notably precipitation. This means that if the forcing data change, the soil parameters that result in reproducing simulated streamflow will change accordingly. Our comparison of hydrological simulations driven by four forcing datasets revealed that PCIC simulations are noticeably better with high NSE values and reproduce hydrographs similar to that from observations. Therefore, we only used PCIC forcing data for VIC model simulations to investigate the uncertainties in the model calibration process. Our primary goal is to evaluate optimizer sensitivity to a unique set of parameter limits. We want to see how the MOCOM optimizer results in different optimized parameters and change the overall simulated hydrograph in the calibration process. Evaluating the same sensitivity for each dataset is beyond the scope of this paper. However, in addition to the PCIC dataset, we have now repeated our methodology with the UW dataset. As expected, the optimized final values of calibration parameters are different for each set of initial parameter range producing three different hydrographs at Fraser River, Hope. We will be adding this detail in our revised manuscript. Thank you again for your thoughtful review and for highlighting these difficult but important questions.*

*Along with the above discussion, all the minor comments will be addressed and incorporated in our revised manuscript.*

**Referee # 2:**

The objectives of this study, which focuses on the modelling of snow hydrology of the Fraser River Basin (FRB) of British Columbia (BC), Canada, using the Variable Infiltration Capacity (VIC) model forced with several high-resolution gridded climate datasets (i.e., ANUSPLIN, NARR, UW, and PCIC), were to comprehensively assess uncertainties related to: (i) driving datasets, (ii) optimization of model parameters, and (iii) model calibration during cool and warm phases of the Pacific Decadal Oscillation (PDO).

This is a very well-written paper that deals with relevant subject matters to the hydrological modelling community. The paper delivers sound results with respect to objective (i), however, I believe those associated with objectives (ii) and (iii) are somewhat incomplete. All the results are discussed with respect to mean values. I believe the strength of the paper would be increased if the interannual variability was discussed as well. Moreover, at the end of the paper, the reader is left with an incomplete take-home message; which begs the following question: how could we use the outcomes of the paper if we were asked to run the model to answer questions related to hydroelectric development or other water resources management issue in the FRB? Something is missing here, like a solid recommendation. Here are a few suggestions I can take from the paper: whenever hydrological modelling is performed one should: (i) analyse the stream flow record with respect to previously identified teleconnection correlations? (ii) perform various calibrations in order to find the optimal set of parameter values? (iii) have several snow elevation bands, may be every 100 m? The readers want a clear take-home message from the authors on how to use their findings. Incidentally, the authors made such a reference in the introduction of the paper, but did follow through in their conclusion (last sentence on p. 4, lines 18-20).

I have made suggestions in the following list of specific comments below on how to fulfill what I perceive as shortcomings. As a side note, I found a bit difficult the exercise of jumping back and forth between the content of the main manuscript and that of the Supplemental File. Perhaps, the introduction of the content of the Supplement File and the content of Section 3.1 could be transformed into a useful technical note.

That being said, I strongly encourage the authors to address these comments as I feel the paper could certainly be a significant contribution to the community.

*We agree with the referee's concerns for objectives (ii) and (iii). We are therefore revising our results and discussion with the set of additional VIC simulations. This is to improve the discussion and the overall take home message of our findings. To address and compare interannual variability of VIC simulations driven by four different datasets, we will be analyzing the simulated runoff coefficient of variation (CV) at the Fraser River, Hope for each simulation. We will also expand the conclusion section of our manuscript with recommendations for applications of hydrological modelling and uncertainties in water resource management.*

*Thank you for providing a list of useful studies related to our work. In our revised manuscript, we will include and relate these studies with our results.*

P. 14, lines 4-6, it is written: « The VIC model calibration is applied to the Fraser River's main stem at Hope, BC and the FRB's major sub-basins, namely the Upper Fraser at Shelley (UF), Stuart (SU), 5 Nautley (NA), Quesnel (QU), Chilko (CH) and Thompson-Nicola (TN) Rivers (Fig. 1a and Supplementary Table 1). »

- Does this mean there is a set of parameter values different for each sub-basin and one for the drainage area between Hope and the outlet of each sub-basin? It is common practice to do so with many distributed models, but I do not see anywhere in the paper the performance of the model with respect to each subbasin outlet. Have I missed something?

***In our manuscript, we focused on the Fraser River main stem at Hope, BC for analysis. However, we have calibrated the VIC model for all the major sub-basins of the FRB using each different forcing datasets and results are already presented in Figure 7 of the manuscript. In our revised version, we will add a new table reporting NSE performance of the model for all the sub-basins.***

P. 14, lines 16-18: to be consistent, should not the five-year spin up period been applied to all gridded datasets; that is from 1979 to 1985; or loop over at least a couple of years until convergence is achieved and then undertake the calibration exercise?

***We agree that the VIC model spin up period should be the same for all four forcing datasets. Except for the NARR driven VIC simulations, the PCIC, UW and ANUSPLIN driven simulations are integrated using a five-year spin up period prior to 1979. In our revised manuscript, we will loop recursively the NARR driven simulation for five years using the year 1979 as the forcing data and will re-calibrate the model to ensure our methodology is consistent with the other three sets of simulations.***

P. 15, lines 4-11: Why using the PCIC gridded dataset for the optimizer uncertainty runs? Why modifying the parameter space? Why not conducting several optimizer uncertainty runs and look at the various local optima? That is what is usually done. By reducing the parameter space, the authors are restricting, in theory, the number of local optima, yet there could still be several of them in this modified space. At the end we get three sets of local optima, yet there are multiple other sets. This is an interesting twist, but:

- Please provide more details on this calibration strategy. The outcome of this work is incomplete as I believe there should have been multiple calibrations for each experiment in order to fully illustrate the equifinality. Please discuss.

***The comparison of hydrological simulations driven by four forcing datasets revealed that PCIC simulations are noticeably better with high NSE values and reproduce hydrographs similar to that from observations. Therefore, we only used PCIC forcing data for VIC model simulations to investigate objective (ii). We agree that in the literature (e.g. Lindenschmidt et al. 2007; Shen et al. 2008; Sudheer et al. 2011, etc.), different methodologies and several optimizer uncertainty runs are used to address overall uncertainty in calibration parameters without modifying the parameter space. However our primary goal is to evaluate optimizer sensitivity to a unique set of parameters limits. We want to see how the MOCOM optimizer***

*results in different optimized parameters and changes the overall simulated hydrograph during the calibration process. One can see from our analysis that within a broad range of parameter limits, only one particular subset of parameter range is optimally converging and produces more reliable calibration results with high NSE values. The overall conclusion of this analysis is that the automated optimizers used to converge calibration parameters still rely on the hydrologist's experience and some manual adjustment of initial calibration parameter ranges. We will be adding this discussion in our revised manuscript. Furthermore, to facilitate concerns from Referee 1, we have now repeated this methodology with the UW driven VIC simulation in addition to PCIC-VIC simulations. We will be updating Table 3 accordingly in our revised manuscript.*

P. 15, lines 12-21: why not six (6) experiments to fully explore the question (2 cool phases x 3 warm phases)?

- This is an interesting experiment, yet I believe it is incomplete as not all possibilities were explored (see Seiller *et al*., 2012).
- Are not there any other highly-impacting teleconnection that affect the basin hydrology (*e.g*., ENSO)?

*We agree with the referee's suggestion. We have now extended our experiments by adding two additional PDO runs (Table A). As per the availability of forcing data, we can only evaluate PDO phases within the 1950-2006 time period. For each calibration experiment in one particular phase of the PDO, the automated MOCOM optimizer is used to optimize calibration parameters. The NSE is calculated for the calibration and validation periods using the daily observed streamflow data for the Fraser River at Hope. While evaluating additional PDO experiments, our conclusion remains the same, i.e. the calibration is biased toward the cold or warm phase of the PDO and therefore it is necessary to avoid PDO phase shifts in the hydrological model calibration and validation process. In essence, it is better to run multiple calibration experiments within the required time period.*

*Table A: PDO runs and their performance metrics (NSE coefficient) in the calibration and validation time period evaluated for Fraser River at Hope, BC.*

|  | Calibration | | Validation | |
|---|---|---|---|---|
|  | *NSE (Time slice)* | *PDO Phase (Flows)* | *NSE (Time slice)* | *PDO Phase (Flows)* |
| *PDO1* | *0.84 (1981-1990)* | *Warm (low flows)* | *0.84 (1991-2001)* | *Warm (low flows)* |
| *PDO2* | *0.84 (1956-1965)* | *Cool (high flows)* | *0.85 (1966-1976)* | *Cool (high flows)* |
| *PDO3* | *0.84 (1967-1976)* | *Cool (high flows)* | *0.79 (1977-1987)* | *Warm (low flows)* |
| *PDO4* | *0.81 (1977-1987)* | *Warm (low flows)* | *0.86 (1967-1976)* | *Cool (high flows)* |
| *PDO5* | *0.84* | *Warm* | *0.82* | *Warm* |

| | (1991-2001) | (low flows) | (1981-1990) | (low flows) |
|---|---|---|---|---|

*The FRB is strongly teleconnected to PDO and ENSO phases. Our VIC implementation has been tested for its response to warm and cool phases of the PDO and ENSO. The model realistically simulates anomalies of runoff, SWE, and snowcover under these climatic conditions (Thorne and Woo, 2011). In Figure B (Figure 7 of supplementary document), the effects of the PDO on annual time series of air temperature, precipitation, SWE$_{melt}$ and runoff is shown.*

*Figures C and D represent the response of VIC simulated snowcover (SC) composite anomalies to warm (Figure C) and cool (Figure D) phases of ENSO. ENSO phases are identified in the observation to estimate composites. Consistent with the findings reported in other studies such as Mantua et al. (1997), Fleming et al. (2007) and Shrestha et al. (2016), the anomalies are below (above) normal in warm (cool) ENSO phases.*

*In our uncertainty analysis, we only focused our attention on the PDO teleconnections as the hydrological model calibration is usually performed over many years (more than 5 to 10 years) rather than a few individual years. As ENSO teleconnections are short term (a season or two), it is not feasible to evaluate its influence on the total calibration period.*

[Figure]

*Figure B: Annual variation of air temperature, precipitation, SWE$_{melt}$ and runoff in cool (blue line) and warm (red line) phases of the PDO. Variables are areally-averaged over all of the FRB's grid points.*

[Figure]

*Figure C: Seasonal composite of precipitation (Pr ~ mm/day) and snowcover (SC ~fraction) anomalies in El Niño years.*

[Figure]

***Figure D: Seasonal composite of precipitation (Pr ~ mm/day) and snowcover (SC ~ fraction) anomalies in La Niña years.***

Given the results introduced in Section 3.1, I think it would have been interesting to have the following calibration parameters: snow-to-rainfall temperature threshold and a vertical temperature gradient.

- Does the VIC model have such parameters? If that is the case, why not calibrating them?

- Is there a publication that has introduced a formal sensitivity analysis of the VIC model?

- Why focusing on soil-related parameters solely and not others as well? In other words, the authors should first introduce to the readers all the model parameters and provide a rationale for their choice.

***Our response to referee 1 is copied here.***

*"We selected these calibration parameters based on the manual calibration experience from previous publications such as by Nijssen et al. (1997), Su et al. (2005), Shi et al. (2008), Kang et al. (2015, 2016) and Islam et al. (2016). The VIC is a physically based hydrologic model and has many (about 20, depending on how the term "parameter" is defined) parameters that must be specified. However, the usual implementation approach involves the calibration of only six soil parameters. Such parameters have the largest effects on the hydrograph shape and are the most sensitive parameters in the water balance components (Nijssen et al. 1997; Su et al. 2005). These parameters must be estimated from observations, via a trial and error procedure that leads to an acceptable match of simulated discharge with observations.*

*For the snow calibration, we have fixed the value of thresholds for maximum (at which snow can fall) and minimum (at which rain can fall) air temperature as 0.5°C and -0.5°C, respectively. These values were adjusted based on the region's climatology and are kept constant for all the simulations in the global control file. Parameters related to the snow albedo are adjusted using the traditional VIC algorithm based on the US Army Corps of Engineers empirical snow albedo decay curves for transitions from snow accumulation to ablation. We will be adding this detail in Section 2.4.1 in our revision process."*

P. 19, lines 12-15: « The lower SWE in the NARR-VIC simulation is probably due to the warmer air temperature during winter and spring (Fig. 2b). Winter temperatures being warmer in the NARR dataset may alter the phase of precipitation partitioning with more rainfall than snowfall, and hence less SWE in the NARR-VIC simulation. »

- This is formulated as a hypothesis, but do not the authors have the input precipitation dataset to validate this hypothesis?

*We have validated this result by a VIC model sensitivity experiment where the air temperature was perturbed by 2°C while keeping the precipitation unchanged. The simulated SWE and runoff decreased (nearly 25% decrease in peak runoff) with 2°C rises in air temperature forcings. The detail of this experiment is already mentioned in the manuscript (in its current version) at Page 22, Lines 15-19. We did not alter the phase of precipitation in this experiment considering that the VIC model automatically partitions the precipitation type (solid versus liquid) during its simulation.*

P.20, line 4: what about a sensitivity analysis of the impact of the number of elevation bands on the output variables of interest? Why ten in other words, why not twenty?

*The current literature focusing the VIC model mostly rely on 1 to 5 elevation bands in model implementations (Haddeland at al, 2002; Sherestha et al. 2012, 2016; Oubeidillah et al. 2014 and many more). Depending on the required research question and the available resources, a suitable number of elevation bands can be generated in the VIC model. In our implementation of the VIC model, we have used 10 elevation bands by utilizing a high resolution digital elevation model. Considering the complex terrain, computational time and 25 km resolution grid, 10 elevation bands are sufficient to investigate elevation dependent changes. Increasing elevation bands to 20 will cost more computation time without contributing much to the elevation dependent changes.*

P.26, lines 17-19: « For each set of calibration experiments, the calibration parameters are different, which affects the air temperature lapse rates and thus evapotranspiration, the formation of the snowpack, and the timing of snowmelt. »

- I thought the temperature lapse rate was not a calibration parameter or is it?

*The temperature lapse rate is not a calibration parameter. In the VIC model, mean grid temperature is lapsed to each elevation band. Precipitation falls as snow or rain depending on the lapsed temperature. The model defines the lapse rate based on the elevation and number of elevations bands during its simulation. We will modify this sentence in our revised version.*

P.28, lines 5-6: « The trend analysis further highlighted uncertainties of the NARR Dataset. »

- Uncertainties in terms of what...I am not sure I follow well the authors here. Please provide additional details.

P.29, line 12: not being familiar with the FRB, I was a bit surprise to learn here that the authors had never mentioned in the description of the basin that there are glaciers in the study area!

*In Alberta and British Columbia, glaciers account for an estimated area of 26,700 km$^2$ (Bolch et al. 2010). There are more than 1000 glaciers in the northern and central Rockies spanning across a total area of 838 km$^2$. However these glaciers contribute relatively little to the annual streamflow to rivers of western Canada. For example, Comeau et al. (2009) studied the glacier contributions to streamflow generation by applying a hydrological model at 9 km resolution to the North and South Saskatchewan Rivers originating in the Canadian Rocky Mountains. They separate the mean annual streamflow contributions from glacier retreat and summer melting of the seasonal snowpack. For the period 1975-1998, they estimate that glacier melt (ice volume losses) made up only 2.0% of mean annual discharge of the Bow River in Calgary.*

*In our study, the dynamics of glaciers cannot be simulated in the current version of the VIC model as glaciers and their dynamics are not included in the model physics. Nonetheless the effects of glaciers may not change our results significantly as we have used the VIC model on ~25 km grid cell resolution (625 km$^2$ area per grid cell) in our study. Furthermore, glaciers cover only 1.5% of the FRB (Shrestha et al. 2012) and provide only a modest contribution to streamflow, primarily in late summer (August/early September). Although glacier dynamics are not simulated in the VIC model there are some cells where there is a perennial snowpack. Year after year this water does not melt out. This is a recognized challenge with the VIC model and is dealt with differently in each implementation, ranging from eliminating these cells from analysis (as in our study) to introducing a simple, conceptual representation of glacier mass balance into VIC, modelled using perennial snow in combination with VIC built in snow routines where a portion of VIC grid cells are identified as glacier cells and used to form a glacier mask (Schnorbus et al. 2011). In this study, we compared those cells with perennial snowpack to Baseline Thematic Mapping (BTM) and found that the glaciating cells match the location of observed glaciers. However there were very few grid cells with a perennial snowpack due to the low resolution and were therefore masked in the analysis. We will explicitly mention this in our revised manuscript.*

*All minor comments will be addressed and incorporated in our revised manuscript.*

*References:*

*Bolch, T., Menounos, B., Wheate. R.: Landsat-based inventory of glaciers in western Canada, 1985-2005. Remote Sensing of Environment 114: 127-137, 2010.*

*Comeau, L. E. L., Pietroniro, A., Demuth, M. N.: Glacier contribution to the North and South Saskatchewan Rivers. Hydrological Processes 23: 2640-2653, 2009.*

*Dodson R., Marks, D.: Daily air temperature interpolated at high spatial resolution over a large mountainous region, Clim. Research, 8:1−20, 1997.*

*Fleming S.W., Whitfield P. H., Moore R. D., Quilty E. J.: Regime dependent streamflow sensitivities to Pacific climate modes cross the Georgia–Puget transboundary ecoregion. Hydrological Processes 21: 3264–3287, 2007.*

*Haddeland, B., Matheussen, V., Lettenmaier, D. P. :Influence of spatial resolution in a macroscale hydrologic model, Water Resources Research, 38 (7), pp. 1124–1133, 2002.*

*Kang, D. H., Gao, H., Shi, X., Islam, S. U. and Déry, S. J.: Impacts of a Rapidly Declining Mountain Snowpack on Streamflow Timing in Canada's Fraser River Basin, Sci. Rep., 6, 19299, doi:10.1038/srep19299, 2016.*

*Kang, D. H., Shi, X., Gao, H. and Déry, S. J.: On the changing contribution of snow to the hydrology of the Fraser River Basin, Canada, J. Hydrometeorol., 15(4), 1344–1365, doi:10.1175/JHM-D-13-0120.1, 2014.*

*Lindenschmidt, K. E., Fleischbein, K., and Baborowski, M.: Structural uncertainty in a river water quality modelling system, Ecol. Model., 204, 289–300, 2007.*

*Mantua N. J., Hare S. R., Zhang Y., Wallace J. M., Francis R. C.: A Pacific Interdecadal Climate Oscillation with impacts on salmon production. Bulletin of the American Meteorological Society 78: 1069–1079, 1997.*

*Nijssen, B., Lettenmaier, D. P., Liang, X., Wetzel, S. W., Wood, E. F.: Streamflow simulation for continental-scale river basins. Water Resour. Res., 33, 711–724, 1997.*

*Oubeidillah, A. A., Kao, S. C., Ashfaq, M., Naz, B. S. and Tootle, G.: A large-scale, high-resolution hydrological model parameter data set for climate change impact assessment for the conterminous US, Hydrol. Earth Syst. Sci., 18(1), 67–84, doi:10.5194/hess-18-67-2014, 2014.*

*Schnorbus, M., Bennett, K., Werner, A. and Berland, A. J.: Hydrologic Impacts of Climate Change in the Peace, Campbell and Columbia Sub-basins, British Columbia, Canada. Pacific Climate Impacts Consortium, University of Victoria: Victoria, BC, 157 pp., 2011*

*Shen, Z. Y., Hong, Q., and Yu, H.: Parameter uncertainty analysis of the non-point source pollution in the Daning River watershed of the Three Gorges Reservoir Region, China, Sci. Total Environ., 405, 195–205, 2008.*

*Shi, X., Wood, A. W. and Lettenmaier, D. P.: How Essential is hydrologic model calibration to seasonal streamflow forecasting?, J. Hydrometeorol., 9(6), 1350–1363, doi:10.1175/2008JHM1001.1, 2008.*

*Shrestha, R. R., Schnorbus, M. A., Peters, D. L.: Assessment of a hydrologic model's reliability in simulating flow regime alterations in a changing climate, Hydrol. Processes, doi: 10.1002/hyp.10812, 2016.*

*Shrestha, R. R., Schnorbus, M. A., Werner, A. T., Berland, A. J.: Modeling spatial and temporal variability of hydrologic impacts of climate change in the Fraser River basin, British Columbia, Canada. Hydrol. Processes, 26, 1840–1860, doi:10.1002/hyp.9283, 2012.*

*Su, F. G., et al.: Streamflow simulations of the terrestrial Arctic domain, J. Geophys. Res.-Atmos., 110(D8), 2005.*

*Sudheer, K. P., Lakshmi, G., and Chaubey, I.: Application of a pseudo simulator to evaluate the sensitivity of parameters in complex watershed models, Environ. Modell. Softw., 26, 135–143, 2011*

*Thorne R. and Woo M. K.: Streamflow response to climatic variability in a complex mountainous environment: Fraser River Basin, British Columbia, Canada, Hydrol. Processes 25, 3076–85, 2011.*

---

## Author Response (AR1)

Environmental Science and Engineering
University of Northern British Columbia
Prince George, BC, Canada
Email: stephen.dery@unbc.ca
Tel: +1 (250) 960-5193
Fax: +1 (250) 960-5845

12 January 2017

Dr. Mark F. P. Bierkens
Special Eric Wood Issue
Hydrology and Earth System Sciences

Dear Dr. Bierkens:

Thank you for your decision on our manuscript titled "Evaluating uncertainties in modelling the snow hydrology of the Fraser River Basin, British Columbia, Canada" by Siraj Ul Islam and Stephen J. Déry (Reference # HESS-2016-469). We fully appreciate the detailed and constructive comments provided by the anonymous Referees #1 and #2. We have carefully revised our manuscript by addressing all concerns and comments of the referees' to the best of our ability. Our point-by-point response (in bold italic lettering) to each of the referees' comment/suggestion is enumerated in the attached document.

Please let us know if additional information or points of clarification are required.

Thank you.

Sincerely yours,

Stephen Déry

**Referee # 1:**

This manuscript provides an assessment of the uncertainties in hydrological modelling over the FRB, BC, Canada. The authors used four precipitation dataset and a calibration routine to quantify and characterize the uncertainties to related to meteorological and parameters estimates. The manuscript is well written and covers an interesting topic, however, I feel some issues have not been properly addressed. I suggest moderate revisions to the manuscript before publication. In general I think some of the procedures could be implemented more consistent to ensure that the conclusions of the paper are better.

*We recognize that there are complexities in our procedures that were not thoroughly addressed. In our revision process, we have paid special attention to ensure our results and discussions are more consistent. This is accomplished by means of additional hydrological model simulations and analyses.*

Page 11 Line 19-20 The authors want to study the impact of different forcing dataset and their related uncertainties in hydrological simulations. They mention a couple of times that the errors in mountainous precipitation might cause significant discrepancies between dataset and therefore they use four different datasets to study that impact. Although the datasets all have their own resolution, the authors decide to bilinear interpolate all of them to a common grid of 25km without taking into account elevation corrections. I believe this procedure will add to the forcing uncertainty, especially for the coarser resolution since they are downscaled to a resolution without including the high resolution elevation data to correct for orographic effects. I think it would be good if the authors could provide some estimate for the uncertainty added by the bilinear interpolation without conditioning on the elevation profile.

*We agree that the elevation correction should be explicitly accounted for in the interpolation methodology to correct for orographic effects. Such correction is highly important when interpolating from coarser to higher spatial resolutions, especially in mountainous terrain (Dodson and Marks, 1997). However, in our interpolation methodology the NARR (32 km) dataset is interpolated from coarse resolution, curvilinear grids to slightly higher (25 km) resolution, rectilinear grids. On the other hand, both the PCIC (6 km) and ANSUPLIN (10 km) datasets are interpolated to a coarser resolution (25 km). Interpolating the NARR dataset from a 32 km to a 25 km resolution does not induce much elevation dependent uncertainties since the change in orography remains minimal between mean elevations at 25 km and 32 km grid resolutions (Figure A). Thus the relationship of atmospheric variables such as air temperature with elevation remains nearly identical at both resolutions. In the revision, we have nonetheless highlighted this source of uncertainty at Page 11, Lines 15-21 and Page 12, Lines 1-2 along with the details of the interpolation process.*

[Figure]

***Figure A: Comparison of mean elevation (m) at 25 km and 32 km resolutions over the part of British Columbia covering the Fraser River Basin (black outlines).***

Section 2.4.1 Why did the authors select these parameters? Is there sensitivity information that could be used to identify the most sensitive parameters? Maybe the impact of the routing model is more prominent, while it is not calibrated and doesn't account for the reservoirs and lakes present in the FRB. In addition, none of the snow parameters is calibrated, while the authors mention the importance of snow throughout the manuscript. Maybe a calibration on the snow processes (compaction, sublimation or just simple degree day factor), would further benefit the discharge simulation at the outlet and for the sub-catchments.

*We selected these parameters for calibration based on the manual calibration experience from previous studies by Nijssen et al. (1997), Su et al. (2005), Shi et al. (2008), Kang et al. (2014, 2016) and Islam et al. (2016), among others. VIC is a physically based hydrologic model that has many (about 20, depending on how the term "parameter" is defined) parameters that must be specified. However, the usual implementation approach involves the calibration of only six soil parameters. Such parameters have the largest effects on the hydrograph shape and are the most sensitive parameters in the water balance components (Nijssen et al. 1997; Su et al. 2005). These parameters must be estimated from observations, via a trial and error procedure that leads to an acceptable match of simulated discharge with observations.*

*For the snow calibration, we have fixed the value of thresholds for maximum (at which snow can fall) and minimum (at which rain can fall) air temperature as 0.5°C and -0.5°C, respectively. These values were adjusted based on the region's climatology and are kept constant for all the simulations in the global control file. Parameters related to the snow albedo are adjusted using the traditional VIC algorithm based on the US Army Corps of Engineers empirical snow albedo decay curves for transitions from snow accumulation to ablation. We have added this detail in Section 2.3.2 at Page 14, Line 20 and Page 15, Lines 1-13 of the revised paper.*

Page 15, why is on the PCIC forcing data used for the optimizer. This almost ensures that the PCIC will have the best performance in the following evaluation sections. It might be more interesting (and more work), to calibrate for every forcing dataset individual and then use these four parameter sets for the validation with every unique forcing dataset leading to sixteen combinations. This also gives you four simulations per forcing dataset as a result of the different calibrations. I know this is some work, but it is feasible. I feel it would lift the quality of the overall uncertainty analysis and thereby better support the conclusions of the paper.

*As discussed in our manuscript, the VIC model is quite sensitive to the meteorological forcing data, most notably precipitation. This means that if the forcing data change, the soil parameters that result in reproducing simulated streamflow will change accordingly. Our comparison of hydrological simulations driven by four forcing datasets revealed that PCIC and UW driven VIC simulations are noticeably better with high NSE values and reproduce hydrographs similar to that from observations. We only used PCIC forcing data for VIC model simulations to investigate the uncertainties in the model calibration process. Our primary goal is to evaluate optimizer sensitivity to a unique set of parameter limits. We want to see how the MOCOM optimizer results in different optimized parameters and change the overall simulated hydrograph in the calibration process. Evaluating the same sensitivity for each dataset is beyond the scope of this paper. However, we have repeated our methodology with the UW dataset with new set of parameters limits. Instead of three, we have run five different experiments. As expected, the optimized final values of calibration parameters are different for each set of initial parameter range producing different hydrographs for the main stem Fraser River at Hope, BC. In the revised manuscript, we only used the UW dataset to force VIC model as this dataset along with our VIC model implementation is examined extensively over the FRB in Kang et al, (2014) and (2016). We have updated the text in section 3.3 of our revised manuscript and have modified Figure 8 and Table 3 accordingly. Thank you again for your thoughtful review and for highlighting these difficult but important questions.*

**Minor comments:**

I have discussed the ANUSPLIN acronym with some colleagues over the lunch break. We believe the authors could maybe come up with a better name.

*This dataset is based on the statistical models generated by the Australian National University Spline (ANUSPLIN), which are the thin-plate smoothing spline-based climate interpolation algorithms (Hutchinson et al., 2009; Hopkinson et al., 2011). In studies such as Eum et al. (2014), Curry et al. (2016) and Irwin et al. (2017), ANSUPLIN is used as the standard name*

*for this dataset. We have therefore retained the same name to remain consistent with common usage in the literature.*

Page 3 Line 13-14: measurement of the response metric -> objective function and the calibration variable

*We have modified this line by replacing "measurement of the response metric" with "objective function and the calibration variable".*

Page 7 Line4: Why did the author select the PDO rather than the more influential ENSO signal for Western Canada.

*We agree that the FRB is strongly teleconnected to both PDO and ENSO phases. However, we only focused our attention on the PDO teleconnections as the hydrological model calibration is usually performed over many years (minimum of 5 to 10 years) rather than a few individual years. As ENSO teleconnections are short term (a season or two), it is not feasible to evaluate its influence on the total calibration period. We have discussed this issue in detail in response to Referee #2.*

Page 11 Line 8: The PGF is not really high temporal resolution, it uses satellite observations, which are corrected with gauges at monthly temporal resolution. I think it would be good to provide the reader with the data sources of the PGF. This is important to understand the performance of the WU dataset.

*Thank you for your suggestion. The monthly precipitation data are sourced from the University of Delaware. The gauge undercatch is first applied on the monthly data and are then interpolated to daily values based on the work by Sheffield et al. (2006). We have extended this discussion by including the following detail at Page 10, Lines 20-21 and Page 11, Lines 1-3 in our revised manuscript.*

*"To improve the precipitation estimates, the monthly data are adjusted to account for gauge undercatch by using the methods outlined by Adam and Lettenmaier (2008). Such adjustment is important since gauge-based precipitation measurements may underestimate solid precipitation in winter by 10%–50% (Adam and Lettenmaier 2003)."*

Section 2.2 and 2.4, maybe combine these sections since they both cover VIC and could be easily combined into on VIC section. Otherwise move section 2.3 forward to have the two VIC sections following one another.

*We have combined VIC model related discussions as section 2.3 in our revised manuscript.*

Page 12 Line 19, maybe remove the number of columns and rows, the domain would be sufficient

*We have modified the sentence at Page 14, Line 2 by removing "composed of 34 rows and 42 columns".*

Page 13 Line 3 Citation, year could be without the brackets

*Correction is made at Page 14, Line 6.*

Page 14 Line 18 Why not loop over the year 1979 rather than no spin-up. If the forcing of 1979 were to be recycled for five year and the stabilized ICs could then be used rather than no spin-up. This would ensure that the NARR simulation is more equal to the others and therefore the difference can be really attributed to the difference in forcing rather than a cold model start.

*We agree that the VIC model spin up period should be the same for all four forcing datasets. Except for the NARR driven VIC simulations, the PCIC, UW and ANUSPLIN driven simulations are integrated using a five-year spin up period prior to 1979. In our revised manuscript, we have looped recursively the NARR driven simulation for five years using the year 1979 as the forcing data and have re-run the model to ensure our methodology is consistent with the other three sets of simulations. However the results and conclusions of the manuscript do not change as there is little difference in the outcome of the NARR simulations. We have revised the text at Page 16, Lines 16-19 accordingly.*

Page 14 Line 18-20 Once calibration. . .(Table 1) -> please clarify. It is not entirely clear what you want to do here.

*We have revised the text as follows at Page 16, Lines 19-20.*

*"After calibration, the model validation runs were initialized with five different state files to produce five ensemble members."*

Page 18 Line 9-12 Do you have estimates of the cross-correlation between the precipitation products. Up to what extend are they derived from the same input data.

*The cross correlation of the ANUSPLIN, UW and PCIC datasets (being driven by the different number of station observations) could be high but the bias in precipitation magnitude makes these datasets different. This is due to the different methodologies used in their development.*

Page 22 Line 15 mentions a VIC sensitivity experiment, it would be great to show some of these results to get a better understanding of the model parameter uncertainty.

*We have now included the SWE and hydrograph of the NARR sensitivity experiments as Figure 5 in the supplementary document.*

Page 24 Line 13-14 "air temperatures are more crucial for hydrological simulations", I would argue that this is true for the timing, but not so much for the total streamflow volume (at least not for the FRB, where evap is low). Maybe rephrase to "are crucial for the runoff timing in hydrological simulations"

*We have modified the text at Page 27, Line 4-5 as suggested.*

Figure 2 When does the water year start? October 1st? Please clarify the captions of the figures, they could be more self-explaining

*Caption of Figure 2 is revised accordingly.*

Figure 7, What is a,b,c,d,e,f? No explanation, also not in the caption of Figure 6

*Caption of Figure 7 is revised as:*

*"Same as Figure 6 but for the FRB's six major sub-basins (a) Fraser-Shelley (UF), (b) Stuart (SU), (c) Nautley (NA), (d) Quesnel (QU), (e) Chilko (CH) and (f) Thompson-Nicola (TN).*

**Referee # 2:**

The objectives of this study, which focuses on the modelling of snow hydrology of the Fraser River Basin (FRB) of British Columbia (BC), Canada, using the Variable Infiltration Capacity (VIC) model forced with several high-resolution gridded climate datasets (i.e., ANUSPLIN, NARR, UW, and PCIC), were to comprehensively assess uncertainties related to: (i) driving datasets, (ii) optimization of model parameters, and (iii) model calibration during cool and warm phases of the Pacific Decadal Oscillation (PDO).

This is a very well-written paper that deals with relevant subject matters to the hydrological modelling community. The paper delivers sound results with respect to objective (i), however, I believe those associated with objectives (ii) and (iii) are somewhat incomplete. All the results are discussed with respect to mean values. I believe the strength of the paper would be increased if the interannual variability was discussed as well. Moreover, at the end of the paper, the reader is left with an incomplete take-home message; which begs the following question: how could we use the outcomes of the paper if we were asked to run the model to answer questions related to hydroelectric development or other water resources management issue in the FRB? Something is missing here, like a solid recommendation. Here are a few suggestions I can take from the paper: whenever hydrological modelling is performed one should: (i) analyse the stream flow record with respect to previously identified teleconnection correlations? (ii) perform various calibrations in order to find the optimal set of parameter values? (iii) have several snow elevation bands, may be every 100 m? The readers want a clear take-home message from the authors on how to use their findings. Incidentally, the authors made such a reference in the introduction of the paper, but did follow through in their conclusion (last sentence on p. 4, lines 18-20).

*We agree with the referee's concerns for objectives (ii) and (iii). We have therefore revised our results and discussion with the set of additional VIC simulations. This is to improve the discussion and the overall take home message of our findings. To address and compare interannual variability of VIC simulations driven by four different datasets, we have analyzed the simulated runoff coefficient of variation (CV) for the Fraser River at Hope, BC for each simulation. We have also expanded the conclusion section of our manuscript with recommendations for applications of hydrological modelling and uncertainties in water resource management.*

I have made suggestions in the following list of specific comments below on how to fulfill what I perceive as shortcomings. As a side note, I found a bit difficult the exercise of jumping back and forth between the content of the main manuscript and that of the Supplemental File. Perhaps, the introduction of the content of the Supplement File and the content of Section 3.1 could be transformed into a useful technical note.

*To limit number of figures, we have only presented core results in the manuscript and have moved all the extra figures to the supplementary document. We expect that the online version of the article in HESS will provide a direct link to the supplementary document to facilate the reader going back and forth between documents.Further to this, it is important the reader has immediate access to the supplementary results as publishing this as a technical note would require considerable time and effort leading to a separate publication, which would inhibit*

*understanding of the current effort. As such, we maintain the use of a supplementary document for this paper.*

That being said, I strongly encourage the authors to address these comments as I feel the paper could certainly be a significant contribution to the community.

**Specific comments:**

To my knowledge several researchers have conducted studies related to objective (i) of this paper and, thus, the authors should consult the following references and relate their work accordingly:

- Essou, G.R.C., R. Arsenault, R., F.P. Brissette. 2016. Comparison of climate datasets for lumped hydrological modeling over the continental United States. J.of Hydrology, 537: 334-345.

- Essou, G.R.C., F. Saberly, P. Lucas-Picher, F.P. Brissette, A. Poulin. 2016. Can Precipitation and Temperature from Meteorological Reanalyses Be Used for Hydrological Modeling? J. of Hydrometeorology, DOI: 10.1175/JHM-D-15-0138.1

- Arsenault, R., F.P. Brissette. 2014. Continuous streamflow prediction in ungauged basins: the effects of equifinality and parameter set selection on uncertainty in regionalization approaches. Water Resour. Res., 50 (7): 6135-6153

- Sabarly, F., G. Essou, P. Lucas-Picher, A. Poulin, F. Brissette, 2016: Use of Four Reanalysis Datasets to Assess the Terrestrial Branch of the Water Cycle over Quebec, Canada. Journal of Hydrometeorology, 17, 1447-1466

***Thank you for providing a list of useful studies related to our work. In our revised manuscript, we have included them and relate these studies to our results.***

To my knowledge, two other studies have dealt with the problem of using calibration and validation series that have different climatological characteristics (objective (iii) of the paper), those of Klemeš (1986. Operational testing of hydrological simulation models. Hydrolog. Sci. Journal 31(1): 13-24) and Seiller et al. (2012. Multimodel evaluation of twenty lumped hydrological models under contrasted climate conditions. Hydrology and Earth System Sciences, European Geosciences Union, 16 (4): 1171-1189.).

- In the introductory section of the manuscript, I invite the authors to relate objective (iii) of the study with the content of the aforementioned papers.

***All these studies are now cited in the introduction at Page 7, Line 10.***

P. 5, line 19: the authors refer to Islam *et al*. (2016), but it is missing from the list of references.

***This citation is now included in the list of references.***

P. 14, lines 4-6, it is written: « The VIC model calibration is applied to the Fraser River's main stem at Hope, BC and the FRB's major sub-basins, namely the Upper Fraser at Shelley (UF),

Stuart (SU), 5 Nautley (NA), Quesnel (QU), Chilko (CH) and Thompson-Nicola (TN) Rivers (Fig. 1a and Supplementary Table 1). »

- Does this mean there is a set of parameter values different for each sub-basin and one for the drainage area between Hope and the outlet of each sub-basin? It is common practice to do so with many distributed models, but I do not see anywhere in the paper the performance of the model with respect to each subbasin outlet. Have I missed something?

*In our manuscript, we focused on the Fraser River's main stem at Hope, BC for analysis. However, we have calibrated the VIC model for all the major sub-basins of the FRB using each different forcing datasets and results are already presented in Figure 7 of the manuscript. In response to this comment, we have added a new table (Table A) reporting NSE scores of the model forced with the PCIC data for all the sub-basins in the supplementary document as Supplementary Table 2.*

*Table A: The Nash–Sutcliffe coefficient of efficiency (NSE) for the PCIC driven VIC calibration (1979–1990) and validation (1991–2006) periods for six major sub-basins of the FRB.*

| Basins | 1979-1990 Calibration | 1991-2006 Validation |
|---|---|---|
| | Daily NSE | Daily NSE |
| Fraser-Shelley (UF) | 0.77 | 0.76 |
| Stuart (SU) | 0.70 | 0.86 |
| Nautley (NA) | 0.23 | 0.15 |
| Quesnel (QU) | 0.89 | 0.83 |
| Chilko (CH) | 0.69 | 0.65 |
| Thompson- Nicola (TN) | 0.86 | 0.80 |

P. 14, lines 16-18: to be consistent, should not the five-year spin up period been applied to all gridded datasets; that is from 1979 to 1985; or loop over at least a couple of years until convergence is achieved and then undertake the calibration exercise?

*We agree that the VIC model spin up period should be the same for all four forcing datasets. Except for the NARR driven VIC simulations, the PCIC, UW and ANUSPLIN driven simulations are integrated using a five-year spin up period prior to 1979. In our revised manuscript, we have looped recursively the NARR driven simulation for five years using the*

*year 1979 as the forcing data and have re-run the model to ensure our methodology is consistent with the other three sets of simulations. We have revised the text at Page 16, Lines 16-19 accordingly.*

P. 15, lines 4-11: Why using the PCIC gridded dataset for the optimizer uncertainty runs? Why modifying the parameter space? Why not conducting several optimizer uncertainty runs and look at the various local optima? That is what is usually done. By reducing the parameter space, the authors are restricting, in theory, the number of local optima, yet there could still be several of them in this modified space. At the end we get three sets of local optima, yet there are multiple other sets. This is an interesting twist, but:

- Please provide more details on this calibration strategy. The outcome of this work is incomplete as I believe there should have been multiple calibrations for each experiment in order to fully illustrate the equifinality. Please discuss.

*The comparison of hydrological simulations driven by four forcing datasets revealed that the PCIC and UW simulations are noticeably better with high NSE values and reproduce hydrographs similar to that from observations. Therefore, we only used PCIC forcing data for VIC model simulations to investigate objective (ii). We agree that in the literature (e.g., Lindenschmidt et al. 2007; Shen et al. 2008; Sudheer et al. 2011, etc.), different methodologies and several optimizer uncertainty runs are used to address overall uncertainty in calibration parameters without modifying the parameter space. However our primary goal is to evaluate optimizer sensitivity to a unique set of parameters limits. We want to see how the MOCOM optimizer results in different optimized parameters and changes the overall simulated hydrograph during the calibration process. One can see from our analysis that within a broad range of parameter limits, only one particular subset of parameter range is optimally converging and produces more reliable calibration results (Figure 8) with high NSE values (Table 3). In the revision process, we have repeated our methodology with the UW dataset with new set of parameters limits. Instead of three, we have run five different experiments. As expected, the optimized final values of calibration parameters are different for each set of initial parameter range producing different hydrographs for the main stem Fraser River at Hope, BC. In the revised manuscript, we only used the UW dataset to force VIC model as this dataset along with our VIC model implementation is examined extensively over the FRB in Kang et al, (2014) and (2016). We have updated the text in section 3.3 of our revised manuscript and have modified Figure 8 and Table 3 accordingly. The overall conclusion of this analysis is that the automated optimizers used to converge calibration parameters still rely on the hydrologist's experience and some manual adjustment of initial calibration parameter ranges. We have added this discussion in our revised manuscript at Page 28, Lines 9-11.*

P. 15, lines 12-21: why not six (6) experiments to fully explore the question (2 cool phases x 3 warm phases)?

- This is an interesting experiment, yet I believe it is incomplete as not all possibilities were explored (see Seiller *et al*., 2012).

- Are not there any other highly-impacting teleconnection that affect the basin hydrology (*e.g*., ENSO)?

*We agree with the referee's suggestion. We have now extended our experiments by adding two additional PDO runs (Table B). As per the availability of forcing data, we can only evaluate PDO phases within the 1950-2006 time period. For each calibration experiment in one particular phase of the PDO, the automated MOCOM optimizer is used to optimize calibration parameters. The NSE is calculated for the calibration and validation periods using the daily observed streamflow data for the Fraser River at Hope, BC. While evaluating additional PDO experiments, our conclusion remains the same, i.e. the calibration is biased toward the cold or warm phase of the PDO and therefore it is necessary to avoid PDO phase shifts in the hydrological model calibration and validation process. In essence, it is better to run multiple calibration experiments within the required time period. We have updated Table 4 and the text at Page 29, Lines 18-20 and Page 30, Lines 1-2 accordingly.*

*Table B: UW driven VIC PDO runs and their performance metrics (NSE coefficient) in the calibration and validation time period evaluated for Fraser River at Hope, BC.*

| | Calibration | | Validation | |
|---|---|---|---|---|
| | *NSE (Time slice)* | *PDO Phase (Flows)* | *NSE (Time slice)* | *PDO Phase (Flows)* |
| *PDO1* | *0.84 (1981-1990)* | *Warm (low flows)* | *0.84 (1991-2001)* | *Warm (low flows)* |
| *PDO2* | *0.84 (1956-1965)* | *Cool (high flows)* | *0.85 (1966-1976)* | *Cool (high flows)* |
| *PDO3* | *0.84 (1967-1976)* | *Cool (high flows)* | *0.79 (1977-1987)* | *Warm (low flows)* |
| *PDO4* | *0.86 (1977-1987)* | *Warm (low flows)* | *0.80 (1967-1976)* | *Cool (high flows)* |
| *PDO5* | *0.89 (1991-2001)* | *Warm (low flows)* | *0.87 (1981-1990)* | *Warm (low flows)* |

*The FRB is strongly teleconnected to PDO and ENSO phases (Thorne and Woo, 2011). Our VIC implementation has been tested for its response to warm and cool phases of the PDO and ENSO. The model realistically simulates anomalies of runoff, SWE, and snowcover under these climatic conditions. In Figure B (Supplementary Figure 8), the effects of the PDO on annual time series of air temperature, precipitation, $SWE_{melt}$ and runoff are shown for the FRB.*

*Figures C and D represent the response of VIC simulated snowcover (SC) composite anomalies to warm (Figure C) and cool (Figure D) phases of ENSO. ENSO phases are identified in the observation to estimate composites. Consistent with the findings reported in other studies such as Mantua et al. (1997), Fleming et al. (2007) and Shrestha et al. (2016), the anomalies are below (above) normal in warm (cool) ENSO phases.*

*In our uncertainty analysis, we only focused our attention on the PDO teleconnections as the hydrological model calibration is usually performed over many years (more than 5 to 10 years)*

*rather than a few individual years. As ENSO teleconnections are short term (a season or two), it is not feasible to evaluate its influence on the total calibration period.*

[Figure]

*Figure B: Annual variation of mean (a) air temperature, (b) precipitation, (c) SWEmelt and (d) runoff in cool (blue line) and warm (red line) phases of the PDO. Air temperature and precipitation are extracted from the UW forcing dataset whereas SWE$_{melt}$ and runoff are the UW-VIC simulation output. (a), (b) and (c) shows areally-averaged values over all FRB's grid cells whereas runoff is calculated using external routing model for Fraser River at Hope.*

[Figure]

*Figure C: Seasonal composite of precipitation (Pr ~ mm day⁻¹) and snowcover (SC ~ fraction) anomalies (1949-2006) in El Niño years. Seasons are based on averaging Dec-Jan-Feb (DJF), Mar-Apr-May (MAM), Jun-Jul-Aug (JJA) and Sep-Oct-Nov (SON) months.*

[Figure]

*Figure D: Seasonal composite of precipitation (Pr ~ mm day⁻¹) and snowcover (SC ~ fraction) anomalies (1949-2006) in La Niña years. Seasons are based on averaging Dec-Jan-Feb (DJF), Mar-Apr-May (MAM), Jun-Jul-Aug (JJA) and Sep-Oct-Nov (SON) months.*

P. 16, lines 2-3: Perhaps the three regions mentioned here should be depicted on the related Figures, that those figures where knowledge of the region is warranted to follow what is presented and discussed in the paper (Figure 4 in the manuscript and Figures 1-3 and 6 in the Supplement File).

***The boundaries of three subregions are already shown in Figure 1b representing the model gridcells and mean elevations. Drawing these boundaries on each spatial panel will render the figures much too crowded. For clarity and easier interpretation of the results, we have therefore not drawn these boundaries on each spatial plot.***

Given the results introduced in Section 3.1, I think it would have been interesting to have the following calibration parameters: snow-to-rainfall temperature threshold and a vertical temperature gradient.

- Does the VIC model have such parameters? If that is the case, why not calibrating them?

- Is there a publication that has introduced a formal sensitivity analysis of the VIC model?

- Why focusing on soil-related parameters solely and not others as well? In other words, the authors should first introduce to the readers all the model parameters and provide a rationale for their choice.

***We selected these parameters for calibration based on the manual calibration experience from previous studies by Nijssen et al. (1997), Su et al. (2005), Shi et al. (2008), Kang et al. (2014, 2016) and Islam et al. (2016), among others. VIC is a physically based hydrologic model that has many (about 20, depending on how the term "parameter" is defined) parameters that must be specified. However, the usual implementation approach involves the calibration of only six soil parameters. Such parameters have the largest effects on the hydrograph shape and are the most sensitive parameters in the water balance components (Nijssen et al. 1997; Su et al. 2005). These parameters must be estimated from observations, via a trial and error procedure that leads to an acceptable match of simulated discharge with observations.***

***For the snow calibration, we have fixed the value of thresholds for maximum (at which snow can fall) and minimum (at which rain can fall) air temperature as 0.5°C and -0.5°C, respectively. These values were adjusted based on the region's climatology and are kept constant for all the simulations in the global control file. Parameters related to the snow albedo are adjusted using the traditional VIC algorithm based on the US Army Corps of Engineers empirical snow albedo decay curves for transitions from snow accumulation to ablation. We have added this detail in Section 2.3.2 at Page 14, Line 20 and Page 15, Lines 1-13 of the revised paper.***

P. 19, lines 12-15: « The lower SWE in the NARR-VIC simulation is probably due to the warmer air temperature during winter and spring (Fig. 2b). Winter temperatures being warmer in the NARR dataset may alter the phase of precipitation partitioning with more rainfall than snowfall, and hence less SWE in the NARR-VIC simulation. »

- This is formulated as a hypothesis, but do not the authors have the input precipitation dataset to validate this hypothesis?

*We have validated this result by a VIC model sensitivity experiment where the air temperature was perturbed by 2°C while keeping the precipitation unchanged. The simulated SWE and runoff decreased (a nearly 25% decrease in peak runoff) with 2°C rises in air temperature forcings. The detail of this experiment is already mentioned in the manuscript at Page 25, Lines 1-4. We did not alter the phase of precipitation in this experiment considering that the VIC model automatically partitions the precipitation type (solid versus liquid) during its simulation. We have now included Supplementary Figure 5 to respresent the result of NARR sensitivity run for daily SWE and discharge.*

P.20, line 4: what about a sensitivity analysis of the impact of the number of elevation bands on the output variables of interest? Why ten in other words, why not twenty?

*The VIC model is mostly configured using 1 to 5 elevation bands in many studies focusing different regions (Haddeland at al. 2002; Shrestha et al. 2012, 2016; Oubeidillah et al. 2014 and many more). Depending on the required research question and the available resources, a suitable number of elevation bands can be generated in the VIC model. In our implementation of the VIC model, we have used 10 elevation bands by utilizing a high resolution digital elevation model. Considering the complex terrain, computational time and 25 km resolution grid, 10 elevation bands are sufficient to investigate elevation dependent changes. Increasing elevation bands to 20 will add substantially more computational time without contributing much to the elevation dependent changes due to model coarse resolution.*

P.22, Section 3.2.2: for each gridded dataset − VIC simulation what about calculating a runoff to precipitation coefficient to make inferences about model robustness?

*The VIC model robustness is thoroughly evaluated in studies such as Kang et al. (2014, 2016) using the UW forcing data for the period 1949-2006 over the FRB and it subbasins. In this manuscript, we are mainly focusing on the comparison of different datasets. Evaluating the runoff to precipitation coefficient is beyond the scope of this paper. However such analysis is already used for evaluating the UW driven VIC simulation over the FRB. For example, Figure E shows the time series of the ratio of water year runoff to precipitation (R/P). There is clearly a declining trend in R/P over the 1949-2006 period, indicating less runoff productivity as the climate warms across the FRB. This is a result of the transition of precipitation from snowfall to rainfall, and a tendency towards greater evapotranspiration.*

[Figure]

*Figure E: Annual ratio of UW driven simulated runoff to precipitation (R/P) in the FRB, water years 1949-2006.*

P.24, line 5: « …(Troin *et al*., 2015, 2016)…» Please check as there is only one Troin *et al*. in the list of reference!

*Reference for Troin et al. (2016) is now included in the reference list as:*

*Troin, M., Poulin, A., Baraer, M. and Brissette, F.: Comparing snow models under current and future climates: Uncertainties and implications for hydrological impact studies, J. Hydrol., 540, 588–602, doi:10.1016/j.jhydrol.2016.06.055, 2016.*

P.26, lines 17-19: « For each set of calibration experiments, the calibration parameters are different, which affects the air temperature lapse rates and thus evapotranspiration, the formation of the snowpack, and the timing of snowmelt. »

- I thought the temperature lapse rate was not a calibration parameter or is it?

*The temperature lapse rate is not a calibration parameter. In the VIC model, mean grid temperature is lapsed to each elevation band. Precipitation falls as snow or rain depending on the lapsed temperature. The model defines the lapse rate based on the elevation and number of elevations bands during its simulation. We have modified this sentence at Page 29, Line 11-12 in our revised manuscript.*

P.28, lines 5-6: « The trend analysis further highlighted uncertainties of the NARR Dataset. »

- Uncertainties in terms of what...I am not sure I follow well the authors here. Please provide additional details.

*We have revised this sentence at Page 30, Lines 19-20 as "The monthly trend analysis distinguished the NARR dataset by showing decreased trends in air temperature and increased trends in precipitation and its VIC driven runoff."*

P29, line 3-4: is not the following sentence « Improving the quality of precipitation data may thus lead to more accurate hydrological responses in the FRB. » contradicting this other sentence « Thus uncertainties in air temperatures are more crucial for hydrological simulations over the FRB rather than those in precipitation» on page , lines 13-14?

- Please help me here!

*Thanks for highlighting this issue. We have revised the first sentence at Page 31, Lines 17-19 as:*

*"While the air temperature plays a dominant role in the hydrological simulations, improving the quality of precipitation data can lead to more accurate hydrological responses in the FRB. Considerable precipitation bias can substantially degrade the model performance."*

P.29, line 12: not being familiar with the FRB, I was a bit surprise to learn here that the authors had never mentioned in the description of the basin that there are glaciers in the study area!

*In Alberta and British Columbia, glaciers account for an estimated area of 26,700 km$^2$ (Bolch et al. 2010). There are more than 1000 glaciers in the northern and central Rockies spanning a total area of 838 km$^2$. However these glaciers contribute relatively little to the annual streamflow to rivers of western Canada. For example, Comeau et al. (2009) studied the glacier contributions to streamflow generation by applying a hydrological model at 9 km resolution to the North and South Saskatchewan Rivers originating in the Canadian Rocky Mountains. They separate the mean annual streamflow contributions from glacier retreat and summer melting of the seasonal snowpack. For the period 1975-1998, they estimate that glacier melt (ice volume losses) made up only 2.0% of mean annual discharge of the Bow River in Calgary.*

*In our study, the dynamics of glaciers cannot be simulated in the current version of the VIC model as glaciers and their dynamics are not included in the model physics. Nonetheless the effects of glaciers may not change our results significantly as we have used the VIC model on ~25 km grid cell resolution (625 km$^2$ area per grid cell) in our study. Furthermore, glaciers cover only 1.5% of the FRB (Shrestha et al. 2012) and provide only a modest contribution to streamflow, primarily in late summer (August/early September). Although glacier dynamics are not simulated in the VIC model there are some cells where there is a perennial snowpack. Year after year this water does not melt out. This is a recognized challenge with the VIC model and is dealt with differently in each implementation, ranging from eliminating these cells from analysis (as in our study) to introducing a simple, conceptual representation of glacier mass balance into VIC, modelled using perennial snow in combination with VIC built in snow routines where a portion of VIC grid cells are identified as glacier cells and used to form a glacier mask (Schnorbus et al. 2011). In this study, we compared those cells with perennial snowpack to Baseline Thematic Mapping (BTM) and found that the glaciating cells match the*

*location of observed glaciers. However there were very few grid cells with a perennial snowpack due to the low resolution and were therefore masked in the analysis. We have included this detail at Page 9, Lines 3-5 and Page 18, Lines 18-20 in the revised manuscript.*

P.29, lines 14-15: « Such structural uncertainties need to be evaluated by examining multiple hydrological models over the same region. »

- That is fine to mention that, but this statement does not have anything to do with what was presented in the paper...why bring this issue and others up at this point, on top of it in the conclusion...maybe there should be a subsection untitled recommendations for future work.

*We have revised the manuscript by removing these lines.*

**Figures:**

Figures 2, 3, 6 & 7

- What about producing a graph illustrating the daily values of the coefficient of variation so the readers can appreciate the interannual variability for each data set?

*As suggested, we have calculated the daily CV of routed runoff for each dataset for the Fraser River at Hope (Figure F). We have added this graph as panel (b) in Figure 6 of the manuscript.*

[Figure]

*Figure F: The observed and simulated daily runoff coefficient of variation (CV) for Fraser at Hope over water years 1979-2006. An external routing model is used to calculate runoff for the ANUSPLIN-VIC, NARR-VIC, UW-VIC and PCIC-VIC simulations.*

Figure 2

- Days with January 1st as Julian day 1? In other words, what are the actual Julian days defining the water Year (1 October to 30 September of the following calendar year)? This comment apply to all relevant graphs introduced in the paper and supplement file.

*The water year is defined in the text at Page 18, Line 16. It starts on 1 October and ends on 30 September of the following calendar year. To assist the reader, we have included the start and end dates of the water year in the figure captions (where required).*

Figure 5

- Boxplots would have been useful to highlight the interannual variability.

*Each panel of Figure 5 represents four different datasets and 10 elevation bands. Including the interannual variability on these plots will make these results more complex by making the different datasets indistinguishable.*

**Other comments**

Other specific comments are all related to verb tense can be found in the annotated manuscript. That is, to improve the reading and understanding I suggest using the past tense instead of the present tense. This may be viewed as an old fashion way of reporting scientific work. Now, I am also open to a strong argument to refute my point of view.

*We have revised the manuscript by using past tense as suggested.*

[revised manuscript text omitted]

---

## Referee Report (RR1)

*Hydrol. Earth Syst. Sci. Discuss.*

Manuscript Number:  doi:10.5194/hess-2016-469, 2016

Title:  Evaluating uncertainties in modelling the snow hydrology of the Fraser River Basin, British Columbia, Canada

**General comments**

This my second review of this paper.  I am satisfied with the answers provided by authors to both sets of reviewers' comments.  Also, I would like to thank the authors for the additional work conducted to improve the overall strength of the results.

I only have two minor comments for the authors that can be found below.

**Specific comments**

- P.12, it is written: « The elevation correction, which is important when interpolating from coarser to higher spatial resolutions (Dodson and Marks, 1997), was not used to correct the orographic effects for the NARR dataset. Interpolating the NARR dataset from a 32 km to a 25 km resolution does not induce much elevation dependent uncertainties since the change in orography remains minimal between mean elevations at 25 km and 32 km grid resolutions. »
    - Please, back up this statement with a quantitative demonstration; that is introduce the supporting numbers.
- P.44, please spell out all authors in the following reference:
    - Su, F. G., et al.: Streamflow simulations of the terrestrial Arctic domain, J. Geophys. Res.-Atmos., 20 110(D8), 2005.

---

## Author Response (AR2)

Environmental Science and Engineering
University of Northern British Columbia
Prince George, BC, Canada
Email: stephen.dery@unbc.ca
Tel: +1 (250) 960-5193
Fax: +1 (250) 960-5845

7 March 2017

Dr. Mark F. P. Bierkens
Special Eric Wood Issue
Hydrology and Earth System Sciences

Dear Dr. Bierkens:

Thank you for your decision on our revised manuscript titled "Evaluating uncertainties in modelling the snow hydrology of the Fraser River Basin, British Columbia, Canada" by Siraj Ul Islam and Stephen J. Déry (Reference # HESS-2016-469). We have now revised our manuscript by addressing the minor comments of the referee. Our responses (in bold italic lettering) to the comments are enumerated in the attached document. Given the nature of these comments, only one minor revision to our manuscript was required.

Please do contact us if you have any questions or concerns in regards to our responses.

Sincerely yours,

Stephen Déry

**Referee:**

General comments

This my second review of this paper. I am satisfied with the answers provided by authors to both sets of reviewers' comments. Also, I would like to thank the authors for the additional work conducted to improve the overall strength of the results.

I only have two minor comments for the authors that can be found below.

*Thank you for thoroughly reviewing our manuscript again. We have now revised the manuscript based on your minor comments.*

Specific comments

• P.12, it is written: « The elevation correction, which is important when interpolating from coarser to higher spatial resolutions (Dodson and Marks, 1997), was not used to correct the orographic effects for the NARR dataset. Interpolating the NARR dataset from a 32 km to a 25 km resolution does not induce much elevation dependent uncertainties since the change in orography remains minimal between mean elevations at 25 km and 32 km grid resolutions. »

> Please, back up this statement with a quantitative demonstration; that is introduce the supporting numbers.

*The grid scale change (%) in elevation between mean elevations at 25 km and 32 km grid resolutions is shown in the Figure R1 below revealing nearly 0% change in orography over most of the region.*

[Figure]

*Figure R1: Spatial change (%) in elevation between 25 km and 32 km resolution.*

*We have now revised the paragraph at Page 11, Lines 18-21 and Page 12, Lines 1-2 as:*

*"The elevation correction, which is important when interpolating from coarser to higher spatial resolutions (Dodson and Marks, 1997), was not used to correct the orographic effects for the NARR dataset. Interpolating the NARR dataset from a 32 km to a 25 km spatial resolution induces negligible elevation dependent uncertainties as elevation changes remain below ±20% in the FRB, with most of the grid cells having nearly no difference in orography."*

• P.44, please spell out all authors in the following reference:

Su, F. G., et al.: Streamflow simulations of the terrestrial Arctic domain, J. Geophys. Res.-Atmos., 20 110(D8), 2005.

*Name of the authors is included in the references as:*

*"Su, F., Adam, J. C., Bowling, L. C. and Lettenmaier, D. P.: Streamflow simulations of the terrestrial Arctic domain, J. Geophys. Res.-Atmos., 110, D08112, doi:10.1029/2004JD005518, 2005".*